# Tyrosine phosphorylation tunes chemical and thermal sensitivity of TRPV2 ion channel

Xiaoyi Mo[1†], Peiyuan Pang[1†], Yulin Wang[1], Dexiang Jiang[1], Mengyu Zhang[1], Yang Li[1], Peiyu Wang[1], Qizhi Geng[1], Chang Xie[1], Hai-Ning Du[1], Bo Zhong[1], Dongdong Li[2], Jing Yao[1,3]*

[1]State Key Laboratory of Virology, College of Life Sciences, Department of Anesthesiology, Zhongnan Hospital of Wuhan University, Frontier Science Center for Immunology and Metabolism, Wuhan University, Wuhan, China; [2]Institute of Biology Paris Seine, Neuroscience Paris Seine, CNRS UMR8246, INSERM U1130, Sorbonne Universite, Paris, France; [3]The Key Laboratory of Neural and Vascular Biology, Ministry of Education Hebei Medical University, Shijiazhuang, China

**Abstract** Transient receptor potential vanilloid 2 (TRPV2) is a multimodal ion channel implicated in diverse physiopathological processes. Its important involvement in immune responses has been suggested such as in the macrophages' phagocytosis process. However, the endogenous signaling cascades controlling the gating of TRPV2 remain to be understood. Here, we report that enhancing tyrosine phosphorylation remarkably alters the chemical and thermal sensitivities of TRPV2 endogenously expressed in rat bone marrow-derived macrophages and dorsal root ganglia (DRG) neurons. We identify that the protein tyrosine kinase JAK1 mediates TRPV2 phosphorylation at the molecular sites Tyr(335), Tyr(471), and Tyr(525). JAK1 phosphorylation is required for maintaining TRPV2 activity and the phagocytic ability of macrophages. We further show that TRPV2 phosphorylation is dynamically balanced by protein tyrosine phosphatase non-receptor type 1 (PTPN1). PTPN1 inhibition increases TRPV2 phosphorylation, further reducing the activation temperature threshold. Our data thus unveil an intrinsic mechanism where the phosphorylation/dephosphorylation dynamic balance sets the basal chemical and thermal sensitivity of TRPV2. Targeting this pathway will aid therapeutic interventions in physiopathological contexts.

*For correspondence:
jyao@whu.edu.cn

†These authors contributed equally to this work

Competing interest: The authors declare that no competing interests exist.

## Editor's evaluation

This important study by Xiaoyi Mo and collaborators identifies and carefully describes a novel mechanism controlling TRPV2 channel sensitivity to heat and 2-APB through phosphorylation/de-phosphorylation of tyrosine residues. The authors identify the specific kinase and phosphatase involved in this mechanism, as well as the specific residues in the channel whose phosphorylation results in channel sensitization to agonists. Further, evidence is provided that this mechanism is physiologically relevant in bone marrow-derived macrophages from rodents.

## Introduction

Transient receptor potential vanilloid 2 (TRPV2) channel is broadly expressed in the body, such as the nervous system (*Caterina et al., 1999*; *Nedungadi et al., 2012*), the immune system (*Link et al., 2010*; *Nagasawa et al., 2007*), and the muscular system (*Peng et al., 2010*; *Zanou et al., 2015*). As a $Ca^{2+}$-permeable polymodal receptor, TRPV2 responds to noxious temperature (>52°C) (*Caterina*

**eLife digest** All the cells in our body have a membrane that separates their interior from the outside environment. However, studded across this barrier are numerous ion channels which allow the cell to sense and react to changes in its surroundings. This includes the ion channel TRPV2, which opens in response to mechanical pressure, certain chemical signals, or rising temperature levels.

Many types of cell express TRPV2, including cells in the nervous system, muscle, and the immune system. However, despite being extensively studied, it is still not clear how TRPV2 opens and closes upon encountering high temperatures. In particular, previous work suggested that TRPV2 only responds when a cell's surroundings reach around 52°C, which is a much higher temperature than cells inside our body normally encounter, even during a fever.

To help resolve this mystery, Mo, Pang et al. studied TRPV2 in neurons responsible for sending sensory information and in immune cells called macrophages which had been extracted from rodents and grown in the laboratory. They found that when the cells were bathed in solutions containing magnesium ions, their TRPV2 channels were more sensitive to a number of different cues, including temperature.

Further biochemical experiments showed that magnesium ions do not directly affect TRPV2, but increase the activity of another protein called JAK1. The magnesium ions caused JAK1 to attach specialized structures called phosphorylation tags to TRPV2. This modification (known as phosphorylation) made the channel more sensitive, allowing it to open in response to temperatures as low as 40°C.

Mo, Pang et al. found that inhibiting JAK1 reduced the activity of TRPV2. Conversely, inhibiting the enzyme that removes the phosphorylation tags, called PTPN1, increased the channel's activity. They also discovered that when JAK1 was blocked, macrophages were less able to 'eat up' bacteria, which is one of their main roles in the immune system.

Taken together these experiments advance our understanding of how TRPV2 becomes active. The balance between the phosphorylation by JAK1 and the dephosphorylation by PTPN1 controls the temperature at which TRPV2 opens. Since TRPV2 contributes to several biological functions, including the development of the nervous system, the maintenance of heart muscles, and inflammation, these findings will be important to scientists in a broad range of fields.

et al., 1999), mechanical force (*McGahon et al., 2016*; *Sugio et al., 2017*), osmotic swelling (*Muraki et al., 2003*), and chemical modulators including 2-aminoethyl diphenylborinate (2-APB) (*Hu et al., 2004*), cannabinoids (*De Petrocellis et al., 2011*), probenecid (*Bang et al., 2007*), tranilast (*Iwata et al., 2020*) and SKF96365 (*Juvin et al., 2007*). TRPV2 has been implicated in diverse biological functions including thermal sensation (*Caterina et al., 1999*), neuronal development (*Shibasaki et al., 2010*), osmotic- or mechanosensation (*Muraki et al., 2003*; *Sugio et al., 2017*), cardiac-structure maintenance (*Katanosaka et al., 2014*), insulin secretion (*Aoyagi et al., 2010*), proinflammatory process (*Entin-Meer et al., 2017*; *Yamashiro et al., 2010*) and oncogenesis (*Siveen et al., 2020*). As TRPV2 knockout mice display normal thermal and mechanical nociception responses (*Park et al., 2011*), whether TRPV2 functions as a temperature sensor or a mechanical sensor in physiology still remains in debate.

The role of TRPV2 in immune responses has also been suggested (*Link et al., 2010*; *Santoni et al., 2013*), such as its regulation of macrophage particle binding and phagocytosis (*Link et al., 2010*). In mast cells, TRPV2-mediated calcium flux stimulates protein kinase A (PKA)-dependent proinflammatory degranulation (*Stokes et al., 2004*). In addition, early studies have shown that peripheral inflammation and phosphoinositide 3-kinase signaling pathways enhance TRPV2 function by recruiting it onto the plasma membrane (*Aoyagi et al., 2010*; *Shimosato et al., 2005*). Meanwhile, the characteristics of TRPV2 activity in endogenous immune cells require to be elucidated.

At the channel level, our recent study found that the lipid-raft-associated protein flotillin-1 interacts with and sustains the surface expression of the TRPV2 channel (*Hu et al., 2021*). The use dependence of the TRPV2 channel in heat sensitivity but not agonist sensitivity has also been reported (*Liu and Qin, 2016*). Recently, the oxidation of TRPV2 on methionine residues was found to activate and sensitize the channel (*Fricke et al., 2019*). Moreover, the structure of TRPV2 at near-atomic resolution has

been determined by cryo-electron microscopy (*Huynh et al., 2016*; *Zubcevic et al., 2016*). Despite the functional and structural insights, the endogenous signaling elements that gate TRPV2 activities remain to be further understood.

Here, we show that the regulator of phosphokinases magnesium (Mg$^{2+}$) exerts an enhancing effect on both the chemical and thermal sensitivity of TRPV2 endogenously expressed in rat bone marrow-derived macrophages (rBMDMs). We then provide evidence that Mg$^{2+}$ activates the phosphokinase JAK1 to increase the phosphorylation levels of TRPV2. In contrast, JAK1 inhibition downregulates TRPV2 channel activity, which in accordance reduces the phagocytic ability of macrophages. We have also determined three JAK1 phosphorylation sites, Y335, Y471, and Y525, in TRPV2. Further, we identify that protein tyrosine phosphatase non-receptor type 1 (PTPN1) is the tyrosine phosphatase that mediates TRPV2 dephosphorylation. Our data unmask an endogenous signaling cascade where tyrosine phosphorylation homeostasis contributes to setting the sensitivity of TRPV2 to thermal and chemical stimuli.

## Results

### Mg$^{2+}$ enhances both the chemical and thermal sensitivity of TRPV2

Enriched in cell cytoplasm, Mg$^{2+}$ regulates the function of a variety of ion channels (*Antonov and Johnson, 1999*; *Cao et al., 2014*; *Lee et al., 2005*; *Luo et al., 2012*; *Obukhov and Nowycky, 2005*). A couple of TRP ion channels have been reported to be modulated by a high concentration of Mg$^{2+}$ (*Cao et al., 2014*; *Yang et al., 2014*). We therefore examined whether TRPV2 activity is sensitive to Mg$^{2+}$. Considering that TRPV2 is abundantly and functionally expressed in macrophages where other types of TRPV channels are barely detectable (*Figure 1—figure supplement 1*; *Link et al., 2010*; *Nagasawa et al., 2007*), we used rBMDMs as an endogenous cell system to record TRPV2 currents. We found that TRPV2 currents at –60 mV evoked by 0.3 mM 2-APB were slowly but dramatically enhanced in the presence of 5 mM Mg$^{2+}$ (*Figure 1A*). The pipette solution contained 1 mM adenosine disodium triphosphate (Na$_2$ATP). In general, Mg$^{2+}$-potentiated responses typically developed over a period of about 100 s to reach a plateau. The presence of 5 mM Mg$^{2+}$ augmented the peak current amplitudes by ~19-fold (*Figure 1B*). Notably, the following response to 0.3 mM 2-APB was somewhat variable but still remained an ~14-fold increase from that before Mg$^{2+}$ treatment (*Figure 1A–B*). We further recorded the effect of Mg$^{2+}$ on TRPV2 current responses in neurons. TRPV2 channels are predominantly expressed in medium- to large-sized dorsal root ganglia (DRGs) neurons that typically express fewer TRPV1 channels (*Caterina et al., 1999*). As illustrated in *Figure 1C–D*, we witnessed similar potentiating effects of Mg$^{2+}$ on 2-APB-evoked currents in a small population of DRG neurons, while the lack of TRPV1 expression was confirmed by the absence of responses to capsaicin, indicating these 2-APB-evoked currents were mediated by TRPV2 channels. To further investigate whether the regulatory effect of Mg$^{2+}$ on TRPV2 reflects a channel-inherent mechanism, we performed recordings in a variety of heterologous expression systems including HEK293T (*Figure 1E–F*), CHO, Hela, and ND7/23 cells (*Figure 1—figure supplement 2*) where TRPV2 was transiently expressed. Indeed, the profound enhancement of TRPV2 activity by Mg$^{2+}$ was observed in all expression cell lines. Additionally, we found that prolonged application of 0.3 mM 2-APB alone didn't have a notable sensitizing effect on TRPV2 currents, while subsequent application of the same stimulus in the presence of 5 mM Mg$^{2+}$ produced a significant increase of the TRPV2 currents, indicating that it was Mg$^{2+}$ not 2-APB that sensitized TRPV2 (*Figure 1—figure supplement 3*).

Next, we asked whether other divalent cations exert similar regulatory effects on TRPV2 currents as Mg$^{2+}$ does. We thus repeated the experiments in TRPV2-expressing HEK293T cells with different cations including Mn$^{2+}$, Ca$^{2+}$, Ba$^{2+}$, Zn$^{2+}$, Cu$^{2+}$, Ni$^{2+}$, Cd$^{2+}$, and Co$^{2+}$. As shown in *Figure 1—figure supplement 4*, among all the tested divalent cations, Ba$^{2+}$, Cu$^{2+}$, and Zn$^{2+}$ had a remarkable inhibition of TRPV2 currents, while Mg$^{2+}$, Mn$^{2+}$, and Co$^{2+}$ enhanced the currents of TRPV2 to different degrees. Among them, Mg$^{2+}$ exhibited a more profound effect on enhancing the TRPV2 channel activity.

To further characterize the regulatory effects of Mg$^{2+}$ on TRPV2 activity, whole-cell currents were elicited by local perfusion of 0.3 mM 2-APB with varied concentrations of Mg$^{2+}$ ranging from 0.1 to 10 mM. Mg$^{2+}$ was effective above 0.1 mM and remained effective up to 20 mM with a half-maximal concentration of 0.94 ± 0.04 mM (*Figure 1G*). In addition, the EC$_{50}$ of 2-APB on TRPV2 activation was shifted to 0.24 ± 0.01 mM from 0.59 ± 0.01 mM in the presence of 5 mM Mg$^{2+}$ (*Figure 1H*).

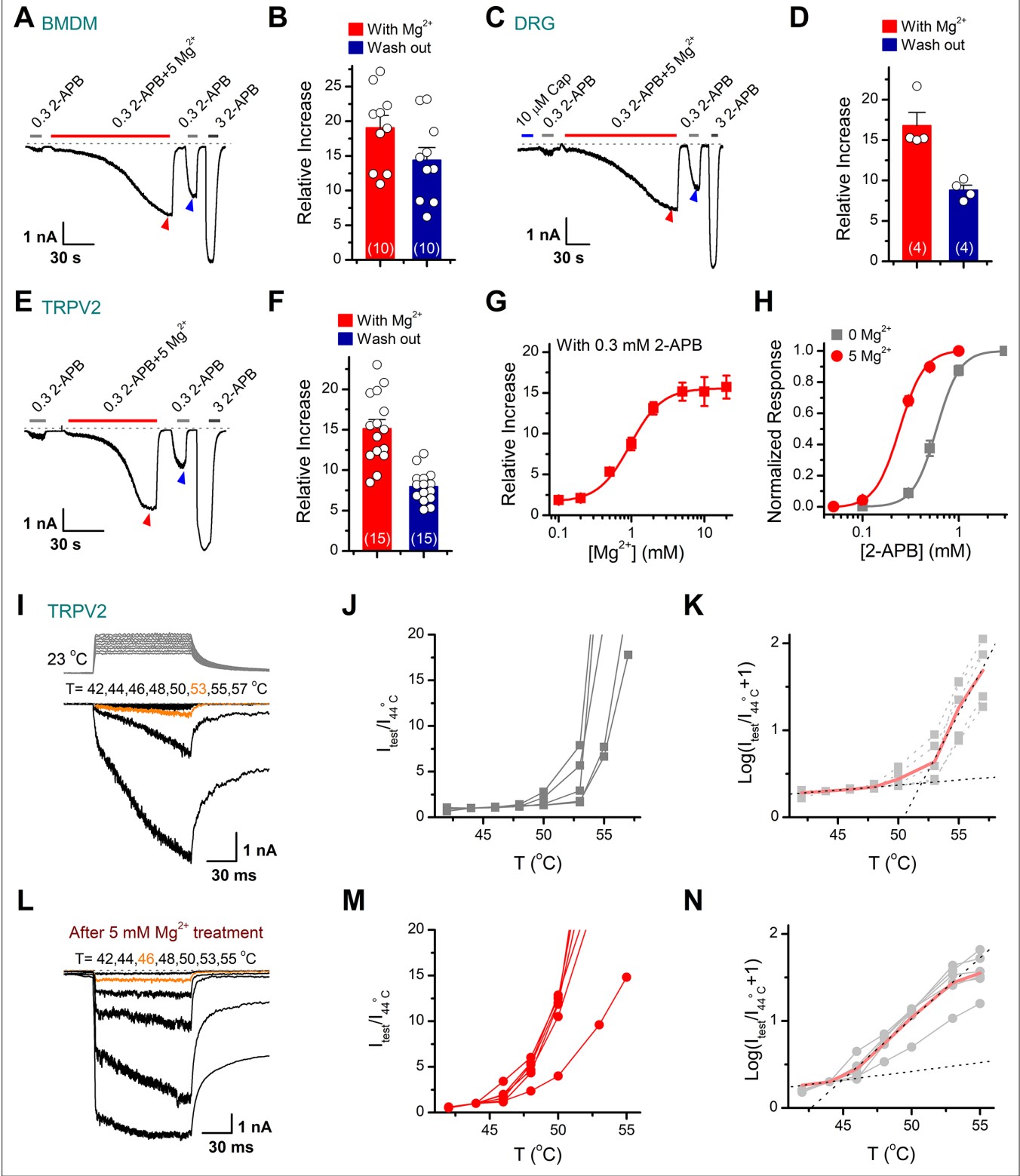

**Figure 1.** Transient receptor potential vanilloid 2 (TRPV2) activities are enhanced in the presence of Mg$^{2+}$. (**A**) Mg$^{2+}$ potentiates 2-aminoethyl diphenylborinate (2-APB) responses in a representative rat bone marrow-derived macrophage (BMDM) cell. The cell was exposed to 0.3 mM 2-APB without or with 5 mM Mg$^{2+}$ and 3 mM 2-APB as indicated by the bars. Membrane currents were recorded in whole-cell configuration, and the holding potential was –60 mV. Bars represent duration of drug application. The dotted line indicates zero current level. (**B**) Summary of relative currents evoked by 0.3 mM 2-APB in the presence of 5 mM Mg$^{2+}$ (indicated by red triangle) and the subsequent application of 0.3 mM 2-APB (indicated by blue triangle).

*Figure 1 continued on next page*

*Figure 1 continued*

Numbers of cells are indicated in parentheses. (**C**) Whole-cell currents at –60 mV in a rat dorsal root ganglion (DRG) neuron treated with 10 µM Cap, 0.3 mM 2-APB, 0.3 mM 2-APB plus 5 mM $Mg^{2+}$, and 3 mM 2-APB. (**D**) Summary of relative currents elicited with 5 mM $Mg^{2+}$ (marked by red triangle) and the subsequent application of 0.3 mM 2-APB (marked by blue triangle). (**E–F**) Parallel whole-cell recordings in TRPV2-expressing HEK293T cells and the relative changes caused by $Mg^{2+}$. (**G**) Dose dependence of $Mg^{2+}$ effects on 2-APB response (0.3 mM). The solid line represents a fit by Hill's equation with $EC_{50} = 0.96 \pm 0.03$ mM and $n_H = 2.0 \pm 0.1$ ($n \geq 5$). (**H**) Dose-response curves of 2-APB for activation of TRPV2 in the presence of 0 or 5 mM $Mg^{2+}$. The solid lines corresponds to Hill's equation with $EC_{50} = 0.59 \pm 0.01$ mM and $n_H = 3.6 \pm 0.1$ for 0 $Mg^{2+}$ ($n = 11$); and $EC_{50} = 0.24 \pm 0.01$ mM and $n_H = 3.4 \pm 0.1$ for application of 5 mM $Mg^{2+}$ ($n = 21$). (**I**) Representative responses to a family of temperature pulses for TRPV2-expressing HEK293T cells under control ($n = 5$). Temperature pulses stepped from room temperature generated by laser irradiation were 100 ms long and had a rise time of 2 ms. (**J**) Current vs. temperature relations at –60 mV obtained from experiments as in (**I**). Individual cells are shown with currents normalized by their amplitude at 44°C. (**K**) Plot of $\log(I_{test}/I_{44^\circ C}+1)$ obtained from the relations in (**I**). (**L–N**) Representative current traces, temperature-activation relations, and plot of $\log(I_{test}/I_{44^\circ C}+1)$ determinations for $Mg^{2+}$ pretreated TRPV2-expressing cells ($n = 6$).

The online version of this article includes the following source data and figure supplement(s) for figure 1:

**Figure supplement 1.** Expression of transient receptor potential vanilloid 2 (TRPV2) in bone marrow-derived macrophages (BMDMs).

**Figure supplement 1—source data 1.** Uncropped, unedited blots for *Figure 1—figure supplement 1B*.

**Figure supplement 1—source data 2.** Uncropped, unedited blots for *Figure 1—figure supplement 1C*.

**Figure supplement 2.** $Mg^{2+}$ potentiates transient receptor potential vanilloid 2 (TRPV2) currents expressed in various cell lines.

**Figure supplement 3.** Effect of $Mg^{2+}$ on 2-aminoethyl diphenylborinate (2-APB)-evoked transient receptor potential vanilloid 2 (TRPV2) currents.

**Figure supplement 4.** Effects of various divalent cations on 2-aminoethyl diphenylborinate (2-APB)-evoked transient receptor potential vanilloid 2 (TRPV2) currents.

**Figure supplement 5.** Effect of intracellular $Mg^{2+}$ on dose responses to 2-aminoethyl diphenylborinate (2-APB) for activation of transient receptor potential vanilloid 2 (TRPV2).

Additionally, we found that the inclusion of 5 mM $Mg^{2+}$ in the pipette solution also increased the TRPV2 channel sensitivity to 2-APB, whereby resulting in a leftward shift of the dose-response curve (*Figure 1—figure supplement 5*).

TRPV2 is a member of the temperature-sensitive ion channel. Therefore, we examined the effect of $Mg^{2+}$ on TRPV2 thermosensitivity using laser irradiation-based temperature controlling and whole-cell recording (*Yao et al., 2009*). HEK293T cells expressing TRPV2 were held at –60 mV when the temperature jumps were delivered (*Figure 1I*, inset). The above experiments showed that the enhanced effect of $Mg^{2+}$ on TRPV2 channel requires long-term continuous treatment, however, prolonged high-temperature stimulation incurs excessive thermal stress and leads to the instability of whole-cell recordings. For such a reason, we first sensitized the TRPV2 channel by stimulating the cells with the combination of 0.3 mM 2-APB and 5 mM $Mg^{2+}$, and then immediately applied the temperature pulses to the same cell right after completely washout 2-APB by bath solution. As illustrated in *Figure 1I–N*, the pretreatment with $Mg^{2+}$ evidently lowered the temperature threshold in TRPV2 activation by ~6°C. Together, these results indicate that $Mg^{2+}$ enhances both the chemical and thermal responses of the TRPV2 ion channel.

## $Mg^{2+}$ potentiates TRPV2 activation via an indirect intracellular pathway

To identify whether $Mg^{2+}$ directly activates TRPV2, we recorded its currents in HEK293T cells using whole-cell patch-clamp in the presence of various concentrations of $Mg^{2+}$ (*Figure 2A*). We observed that even 100 mM $Mg^{2+}$ did not induce any detectable current (*Figure 2A–B*), indicating that extracellular $Mg^{2+}$ cannot directly activate TRPV2 channels. Likely, $Mg^{2+}$ enhances TRPV2 activation via an intracellular mechanism. Thus, extracellularly applied $Mg^{2+}$ might need to permeate into cell cytosol through the activated channel. To probe the mechanism of $Mg^{2+}$-mediated enhancement of TRPV2 activity, we added a high concentration of chelator (EDTA, 20 mM) into the pipette solution to maintain a lower concentration of free intracellular $Mg^{2+}$. As shown in *Figure 2C–D*, chelating intracellular $Mg^{2+}$ with 20 mM EDTA delivered through patch pipette abolished the enhancement effect.

The above results suggest that the enhancing effect of $Mg^{2+}$ on TRPV2 activation takes place on the intracellular side. We then performed inside-out patch-clamp to examine whether $Mg^{2+}$ directly activates TRPV2 from the intracellular side (*Figure 2E*). Akin to extracellular application, even 100 mM $Mg^{2+}$ did not induce any detectable current from the intracellular side (*Figure 2F*). Together, our

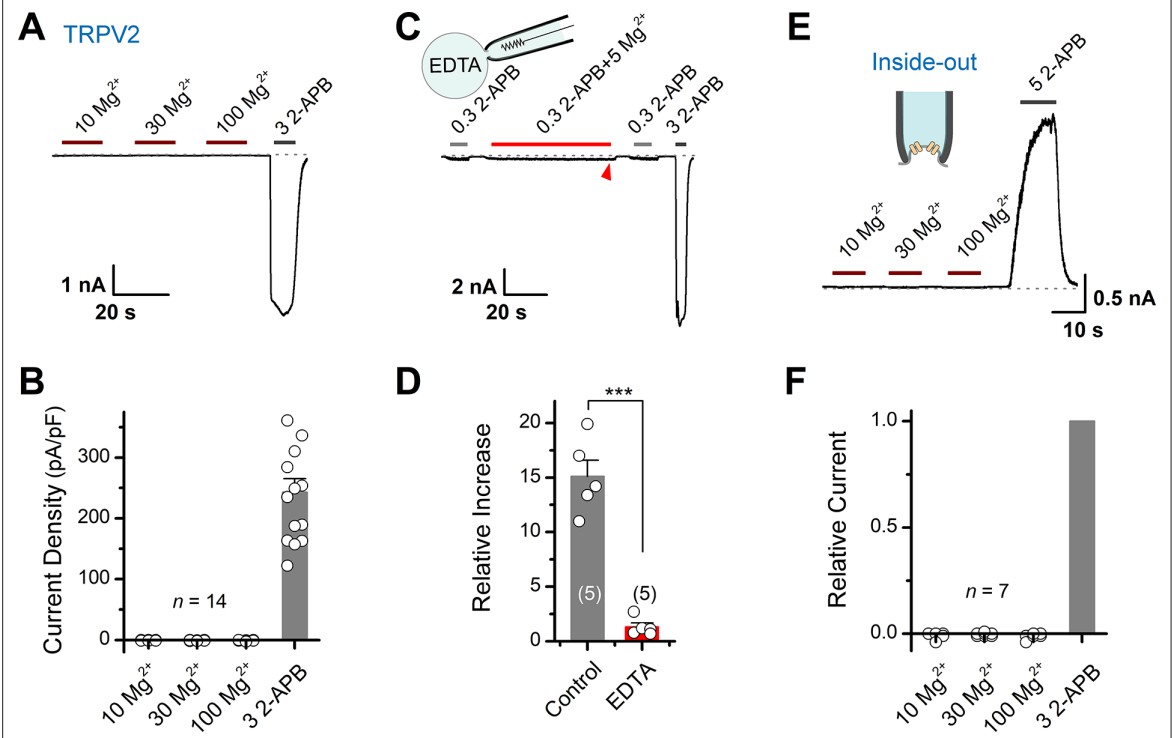

**Figure 2.** $Mg^{2+}$ has an indirect effect on transient receptor potential vanilloid 2 (TRPV2) channels. (**A**) High concentrations of $Mg^{2+}$ have no direct effect on TRPV2 channels from the extracellular side. Representative whole-cell currents at –60 mV in a TRPV2-expressing HEK293T cells consecutively treated with 10, 30, 100 mM $Mg^{2+}$ and 3 mM 2-aminoethyl diphenylborinate (2-APB). (**B**) Comparison of current density evoked by different concentrations of $Mg^{2+}$ and 3 mM 2-APB. (**C**) Whole-cell recordings from TRPV2-expressing HEK293T cells showing the response to 0.3 mM 2-APB, 0.3 mM 2-APB plus 5 mM $Mg^{2+}$, and 3 mM 2-APB. Note the pipette solution contained 20 mM EDTA. (**D**) Average plot of the relative changes. p = 0.0006 by unpaired Student's $t$-test. (**E**) Current traces recorded in inside-out configuration evoked by different concentrations of $Mg^{2+}$ and 5 mM 2-APB. (**F**) Summary plot of relative currents elicited with 10, 30, 100 mM $Mg^{2+}$ and 3 mM 2-APB. The dotted line indicates zero current level.

results suggest that the potentiation effect of $Mg^{2+}$ on TRPV2 activation relies on an indirect intracellular mechanism.

## JAK1-mediated tyrosine phosphorylation regulates TRPV2 sensitivity

Previous studies suggest that some stimuli, like insulin, recruit TRPV2 to the plasma membrane to increase the whole-cell response (*Hisanaga et al., 2009*; *Kanzaki et al., 1999*; *Nagasawa et al., 2007*). To verify whether $Mg^{2+}$ solicits similar mechanisms, we compared the saturation currents evoked by a high dose of 2-APB (3 mM) before and after $Mg^{2+}$ treatment. Our data displayed that subsequent to $Mg^{2+}$ application, though the currents evoked by sub-saturation doses of 2-APB were well potentiated, there was no significant change in the maximum saturation currents (*Figure 3A*). This observation indicates that $Mg^{2+}$ does not alter the expression level of TRPV2 at the plasma membrane.

Alternatively, $Mg^{2+}$ is known as an essential cofactor for enzymatic reactions (*de Baaij et al., 2015*). Especially, $Mg^{2+}$ is an important regulator of phosphokinases and plays a crucial role in their catalytic activity. Enzymatic/catalytic processes also corroborate the fact that the enhancing effect of $Mg^{2+}$ on TRPV2 took a relatively long time (~100 s) and could not be immediately eluted (*Figure 1A–B*). Hence, we hypothesize that $Mg^{2+}$ regulates TRPV2 channels through phosphorylation or dephosphorylation. To test this hypothesis, we investigated the phosphorylation level of immunoprecipitated TRPV2 with anti-phosphotyrosine and anti-phospho-Ser/Thr antibody in the presence of 2-APB agonist, with and without $Mg^{2+}$ (*Figure 3B*). The results revealed a significant increase in tyrosine phosphorylation and serine/threonine phosphorylation levels of TRPV2 in the presence of $Mg^{2+}$. Since the mechanism of phosphorylation involves the transfer of a phosphate (Pi) from ATP to the substrate, we thus used AMP-PNP, a nonhydrolyzable analog of ATP, to replace ATP to inhibit the process of phosphorylation. As shown in *Figure 3C*, the enhancement effect of $Mg^{2+}$ on TRPV2 currents was abolished when

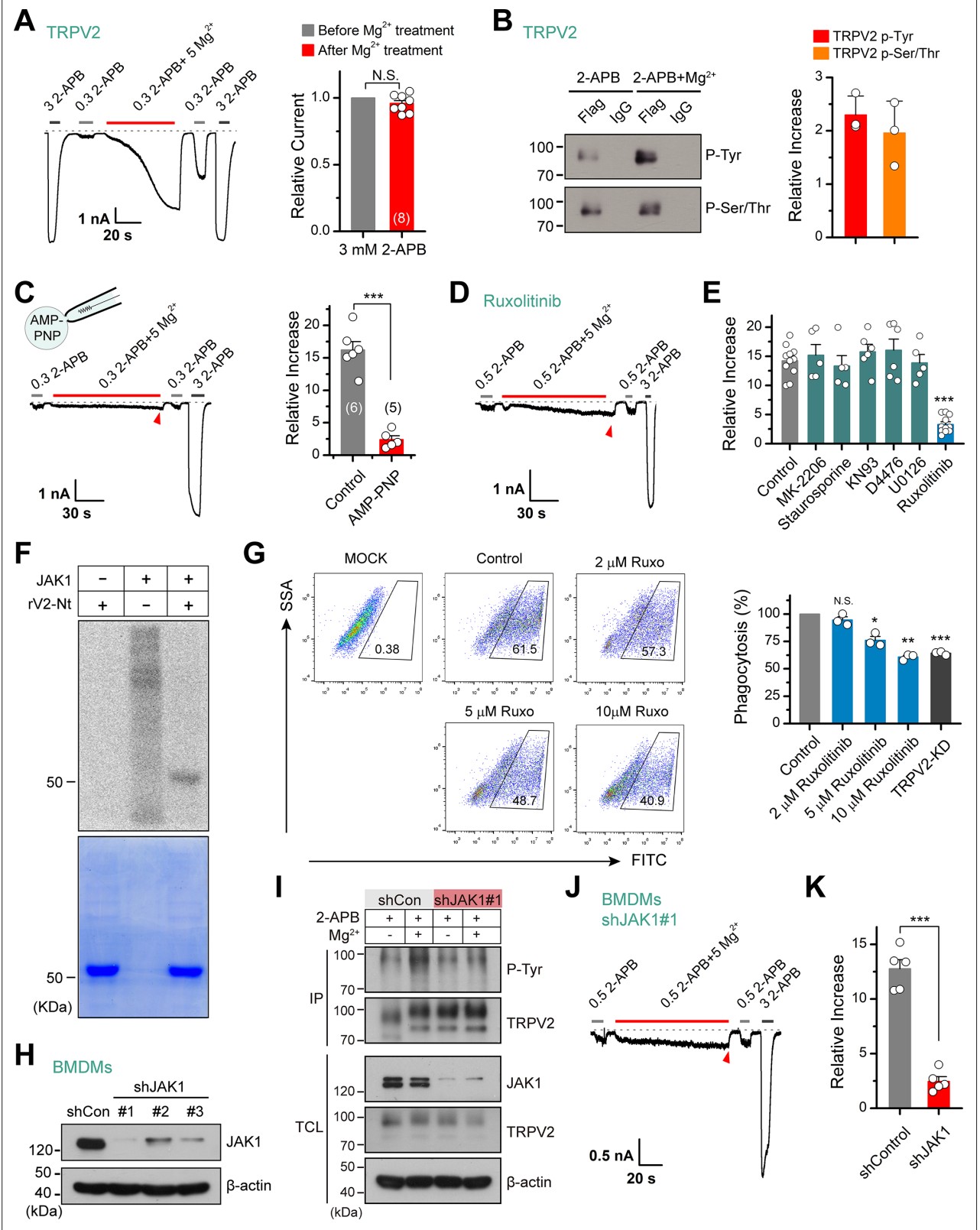

**Figure 3.** Tyrosine phosphokinase JAK1 upregulates channel activity via phosphorylation of transient receptor potential vanilloid 2 (TRPV2). (**A**) Representative whole-cell recordings from TRPV2-expressing HEK293T cells showing the responses to 3 mM 2-aminoethyl diphenylborinate (2-APB) before and after the treatment by 0.3 mM 2-APB plus 5 mM $Mg^{2+}$ (*left*). Average peak responses to 3 mM 2-APB before and after $Mg^{2+}$ application (*right*). The dotted line indicates zero current level. The holding potential was –60 mV. p = 0.12 by one-sample *t*-test. (**B**) Tyrosine phosphorylation

*Figure 3 continued on next page*

*Figure 3 continued*

and serine/threonine phosphorylation of immunoprecipitated TRPV2-Flag transiently transfected in HEK293T cells in the absence and presence of 5 mM $Mg^{2+}$ were determined by immunoblotting with anti-phosphotyrosine antibody (pTyr) and anti-Phospho-(Ser/Thr) Phe antibody (pSer/Thr). *Inset*, Protein amounts of tyrosine-phosphorylated or serine/threonine-phosphorylated immunoprecipitated TRPV2 proteins were quantified, and phospho-Tyr TRPV2/total TRPV2 and phospho-Ser/Thr TRPV2/total TRPV2 were calculated from at least three independent experiments. Error bars indicate SD. (**C**) *Left*, representative whole-cell currents at –60 mV in a TRPV2–expressing HEK293T cell treated with 0.3 mM 2-APB, 0.3 mM 2-APB plus 5 mM $Mg^{2+}$ and 3 mM 2-APB. The pipette solution contained adenosine triphosphate ( ATP) nonhydrolyzable analog adenylyl imidodiphosphate (AMP-PNP). *Right*, summary of relative changes under different conditions. p = 9.29E-6 by unpaired Student's *t*-test. (**D**) Whole-cell currents in response to 2-APB under inhibition of JAK1 by Ruxolitinib. (**E**) Summary plot of $Mg^{2+}$ effects on TRPV2 currents under the various conditions. \*\*\*p < 0.001. (**F**) In vitro kinase assay with [$^{32}$P]-γ-ATP, tyrosine kinase JAK1, and recombinant His-tagged rat TRPV2 N-terminus. Phosphorylation signals were detected by autoradiography. Loading amount of different TRPV2 proteins was accessed by coomassie blue staining. (**G**) Flow cytometry analysis for phagocytosis. Flow cytometry analysis was employed to determine the phagocytosed level of green fluorescent protein (GFP)-expressing *Escherichia coli* (GFP *E. coli*) by bone marrow-derived macrophages (BMDMs) treated with varying concentrations of Ruxolitinib or transfected with shTRPV2#1. Bar graph displaying the effects on phagocytosis under different conditions. \*p < 0.05, \*\*p < 0.01, \*\*\*p < 0.001. (**H**) Immunoblot analysis (with anti-JAK1 or anti-β-actin) of BMDM cells transfected for 72 hr with JAK-1-targeting shRNA (shJAK1#1, shJAK1#2, and shJAK1#3) or shControl to test knockdown efficiency of shRNA. (**I**) Immunoblot analysis of the tyrosine phosphorylation levels of TRPV2 in BMDM cells transfected with shJAK1#3 or shControl for 72 hr in the absence and presence of $Mg^{2+}$, respectively. (**J**) Whole-cell recordings in BMDM cells transfected with shJAK1#3 showing the responses to 0.3 mM 2-APB, 0.3 mM 2-APB plus 5 mM $Mg^{2+}$ and 3 mM 2-APB. (**K**) Comparison of relative increase under different conditions. p = 4.49E-6 by unpaired Student's *t*-test. Error bars indicate standard error of the mean (SEM).

The online version of this article includes the following source data and figure supplement(s) for figure 3:

**Source data 1.** Uncropped, unedited blots for *Figure 3B*.

**Source data 2.** Uncropped, unedited blots and gels for *Figure 3F*.

**Source data 3.** Uncropped, unedited blots for *Figure 3H*.

**Source data 4.** Uncropped, unedited blots for *Figure 3I*.

**Figure supplement 1.** Effects of $Mg^{2+}$ on transient receptor potential vanilloid 2 (TRPV2) responses with or without addition of adenosine triphosphate (ATP) in the pipette solutions.

**Figure supplement 2.** Liquid chromatography-tandem mass spectrometry (LC-MS/MS) analysis of the phosphorylation of transient receptor potential vanilloid 2 (TRPV2).

dialyzed AMP-PNP (4 mM) into the cell through recording pipette, suggesting that $Mg^{2+}$ potentiates phosphorylation of TRPV2 upon agonist stimulation. Interestingly, the sensitizing effect of $Mg^{2+}$ was also observed in the whole-cell recordings without the addition of ATP in the pipette solution (*Figure 3—figure supplement 1*). This is most likely due to the abundance of ATP in the cell which is not rapidly diluted by the patch pipette solutions. Or the intracellular ATP has a concentration gradient at various sites and is associated with endogenous enzymes at different localizations.

We next screened the potential kinases involved by treating the cells with various protein kinase inhibitors. As shown in *Figure 3D–E*, treatment with Ruxolitinib (JAK1 inhibitor) but not MK-2206 (Akt inhibitor), staurosporine (PKC inhibitor), KN93 (CaMKII inhibitor), D4476 (CK1 inhibitor), or U0126 (MEK1/2 inhibitor) abolished the enhancement of TRPV2 activity by $Mg^{2+}$, suggesting that JAK1 is probably the kinase promoting TRPV2 activity.

Utilizing mass spectrometry, we found peptides phosphorylated at the Y335 site that locates on the N terminus (Nt) of TRPV2 (*Figure 3—figure supplement 2*). We next tested whether JAK1 directly phosphorylated TRPV2. Based on this finding and considering the difficulty of the purification of the TRPV2 transmembrane region, we purified TRPV2-Nt for in vitro phosphorylation experiments. Using in vitro kinase assay, we observed that JAK1 directly phosphorylated TRPV2-Nt (*Figure 3F*).

TRPV2 ion channel has been shown to regulate the phagocytosis of macrophages (*Link et al., 2010*). We therefore examined macrophage phagocytosis of GFP-expressing *Escherichia coli* (GFP *E. coli*) using flow cytometry by regulating the activity of TRPV2. As expected, knockdown of TRPV2 by shTRPV2#1 significantly inhibited phagocytosis by BMDM cells (36% ± 1% reduction, n = 3) (*Figure 3G*). We then explored whether inhibition of tyrosine phosphorylation by Ruxolitinib affects BMDM phagocytosis. Indeed, Ruxolitinib reduced macrophage phagocytosis in a concentration-dependent manner, with a reduction of 39% ± 2% observed with 10 µM Ruxolitinib (n = 3). This result thus corroborates the role of phosphorylation in the functional facilitation of TRPV2 activity.

Next, we evaluated the regulatory effect of JAK1 on TRPV2 function using shRNA-mediated knockdown (*Figure 3H*). We observed that selective knockdown of JAK1 expression largely reduced

$Mg^{2+}$-mediated tyrosine phosphorylation of TRPV2 protein (**Figure 3I**). Consistently, knockdown of JAK1 expression inhibited the enhancing effect of $Mg^{2+}$ on TRPV2 current responses in BMDM cells (**Figure 3J–K**). These results together suggest that JAK1 is the kinase underlying $Mg^{2+}$-induced enhancement of TRPV2 activation.

## JAK1 phosphorylates TRPV2 at Y335, Y471, and Y525 molecular sites

Our above results showed that the influx of $Mg^{2+}$ through TRPV2 channel would activate JAK1 and increase the phosphorylation level of the channel, we then investigated the molecular mechanism. Since our mass spectrometry experiment had shown that Y335 was a potential site that may be phosphorylated by JAK1 (**Figure 3—figure supplement 1**), we asked whether the mutation at this site would affect the effect of $Mg^{2+}$ on TRPV2 currents. Indeed, mutating Y335 into phenylalanine to simulate dephosphorylation partially inhibited the enhancement of TRPV2 currents by $Mg^{2+}$ (**Figure 4A–B**). For comparison, the treatment with 5 mM $Mg^{2+}$ increased the 2-APB response (0.3 mM) by approximately 9-fold for mutation Y335F, whereas approximately 16-fold for wild-type (WT) TRPV2. The substitution of Y by F approximates a tyrosine that cannot be phosphorylated, while mutations to the negative charge of aspartic acid (D) or glutamic acid (E) are commonly used to mimic phosphorylated tyrosine (**Pearlman et al., 2011**). As expected, we observed that mutants TRPV2-Y335D and TRPV2-Y335E increased the sensitivity to 2-APB (**Figure 4C–D**). We thus further verified the effect of Y335F mutation on protein phosphorylation status. **Figure 4E** illustrates that JAK1-mediated phosphorylation of TRPV2-Nt was abolished by TRPV2(Y335F) and significantly inhibited by the dominant-negative mutant of JAK1 (JAK1-K908A). These data suggest that Y335 is a critical site for JAK1-mediated tyrosine phosphorylation.

Since mutation Y335F partially abolishes the enhancement effect of $Mg^{2+}$, there may exist other phosphorylation sites in TRPV2 channel protein. Using mutant Y335F as a template, we further mutated the tyrosine residues in the N-terminal ankyrin repeat domain, the membrane-proximal domain, intracellular linkers (Linker), and the C-terminal (Ct) into phenylalanine by site-directed mutagenesis, respectively. We obtained the following mutants: 8YF (Y98/105/111/162/208/228/271/335F), 3YF (Y323/335/343F), 6YF (Y335/455/471/514/515/525F), and 2YF (Y335/675F) (**Figure 4F**). Mutant 6YF greatly reduced the $Mg^{2+}$-induced enhancement of TRPV2 response (**Figure 4G**). When phenylalanine at positions 471 and 525 were reversed back to tyrosine from the 6YF mutant (6YF471Y and 6YF525Y), the enhancement of TRPV2 was rescued (**Figure 4H**).

Triple mutant TRPV2(Y335/471/525F) was generated to confirm the significance of these three specific sites. The results in **Figure 4I–J** displayed that TRPV2(Y335/471/525F) largely eliminated the enhancement of TRPV2 by $Mg^{2+}$. Notably, TRPV2(Y335/471/525F) was a little more sensitive to 2-APB. One possible reason is that the triple mutation might somehow alter the channel conformation and result in the increased sensitivity to its chemical agonist. The protein sequence alignment showed that Y335, Y471, and Y525 amino acid residues are highly conserved in various mammalian TRPV2 homologs (**Figure 4—figure supplement 1**). Moreover, this tri-mutant also downregulated tyrosine phosphorylation levels of immunoprecipitated TRPV2 protein (**Figure 4K**).

To discern the potentiation is depended on intracellular signaling, we repeated the experiments in excised membrane patches. As shown in **Figure 4—figure supplement 2A**, the inside-out recordings from TRPV2-expressing HEK293T cells at +60 mV show that the presence of $Mg^{2+}$ increased the 2-APB response. However, whether the excised membrane patches might attach portion of tyrosine kinase JAK1 remained unknown. Therefore, we further conducted the experiments by pretreatment with the JAK1 inhibitor, Ruxolitinib, which indeed reduced the enhancement caused by $Mg^{2+}$ (**Figure 4—figure supplement 2B**). This confirms the $Mg^{2+}$-induced TRPV2 current enhancement is modulated by JAK1 phosphorylation (**Figure 3**). Moreover, we performed inside-out recordings to test the effect of $Mg^{2+}$ on TRPV2(Y335/471/525F) mutant channel that loses the capability to be phosphorylated by JAK1. As expected, $Mg^{2+}$ failed to enhance the 2-APB-evoked currents (**Figure 4—figure supplement 2C**). Together, these data corroborate that $Mg^{2+}$-JAK1-mediated phosphorylation contributes to the increased sensitivity of the TRPV2 channel (**Figure 4—figure supplement 2D**). Of note, our findings also imply that the excised membrane patches cannot completely isolate the regulatory effect of the intracellular signaling pathway occurring underneath the cell membrane site.

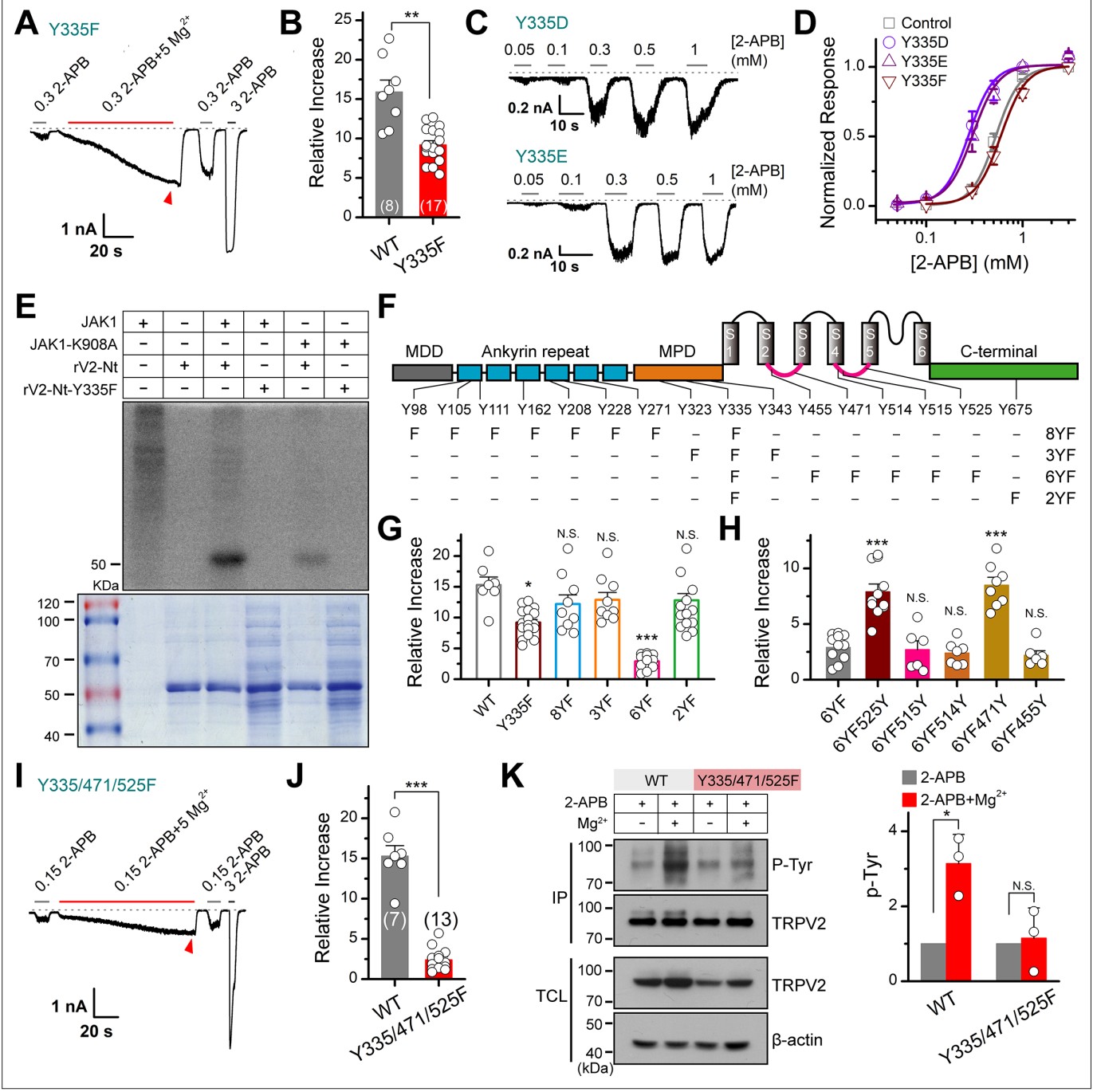

**Figure 4.** JAK1 has three phosphorylation sites on the transient receptor potential vanilloid 2 (TRPV2) channel. (**A**) Representative whole-cell currents at –60 mV elicited with 0.3 mM 2-APB, 0.3 mM 2-APB plus 5 mM $Mg^{2+}$ and 3 mM 2-APB in HEK293T cells that expressed TRPV2(Y335F). Bars represent duration of stimuli. The dotted line indicates zero current level. (**B**) Comparison of relative changes between wild-type TRPV2 and TRPV2(Y335F) following the treatment by $Mg^{2+}$. p = 0.003 by unpaired Student's *t*-test. (**C**) Representative whole-cell currents at –60 mV evoked by varying concentrations of 2-APB in HEK293T cells that expressed TRPV2(Y335D) or TRPV2(Y335E). (**D**) Concentration-response curves of 2-APB for TRPV2 mutants. Solid lines represent fits by a Hill's equation with $EC_{50} = 0.53 \pm 0.01$ mM and $n_H = 3.5 \pm 0.1$ for TRPV2-WT (n = 9); $EC_{50} = 0.28 \pm 0.01$ mM and $n_H = 3.4 \pm 0.2$ for Y335D (n = 8); $EC_{50} = 0.31 \pm 0.01$ mM and $n_H = 3.3 \pm 0.1$ for Y335E (n = 7) and $EC_{50} = 0.60 \pm 0.01$ mM and $n_H = 3.4 \pm 0.2$ for Y335F (n = 8). (**E**) In vitro kinase assay with [$^{32}$P]-γ-ATP, immunoprecipitated tyrosine kinase JAK1 and recombinant His-tagged wild-type or mutant TRPV2 N-terminus. Phosphorylation signals were examined by autoradiography. (**F**) Linear diagram of the TRPV2 channel topology, with all intracellular tyrosine residues labeled, and a summary of substitutions of tyrosine by phenylalanine used in this study. (**G**) Summary plot of the $Mg^{2+}$-dependent enhancement in various mutants. All the TRPV2 mutants retained their normal responses to 2-APB. *p < 0.05, ***p < 0.001. (**H**) Statistic results for the $Mg^{2+}$-dependent enhancement for mutants which were respectively reverse mutated from TRPV2-6YF. ***p < 0.001. (**I**) Representative whole-cell currents at –60 mV

*Figure 4 continued on next page*

*Figure 4 continued*

elicited with 0.15 mM 2-APB, 0.15 mM 2-APB plus 5 mM $Mg^{2+}$ and 3 mM 2-APB in HEK293T cells that expressed TRPV2-Y335/471/525F. (**J**) Average plot of the relative changes of wild-type and Y335/471/525F currents following treatment by $Mg^{2+}$. p = 2.30E-9 0.001 by unpaired Student's *t*-test. (**K**) Immunoblotting analysis with anti-phosphotyrosine antibody (pTyr) showing the tyrosine phosphorylation levels in HEK293T cells transfected with TRPV2 or TRPV2-Y335/471/525F in the absence and presence of $Mg^{2+}$. *Right*, quantitative analysis of the fold increase of tyrosine-phosphorylated TRPV2 proteins and TRPV2(Y335/471/525F) proteins following different treatments (n = 3; means ± SD [standard deviation]). Error bars indicate standard error of the mean (SEM).

The online version of this article includes the following source data and figure supplement(s) for figure 4:

**Source data 1.** Uncropped, unedited blots and gels for *Figure 4E*.

**Source data 2.** Uncropped, unedited blots for *Figure 4K*.

**Figure supplement 1.** Partial amino acid sequence alignment of transient receptor potential vanilloid 2 (TRPV2) channels.

**Figure supplement 2.** Inside-out recordings showing the effects of $Mg^{2+}$ on 2-aminoethyl diphenylborinate (2-APB)-evoked currents.

## Tyrosine phosphorylation enhances chemical and thermal sensitization of TRPV2

Protein phosphorylation is a reversible post-translational modification mediated by kinases and phosphatases. Having characterized JAK1 as the kinase for tyrosine phosphorylation of TRPV2, we next sought to identify the phosphatases that counteracted this process. We took advantage of various protein phosphatase inhibitors to search for the phosphatases that mediated the dephosphorylation of TRPV2. The protein phosphatases comprise the phosphoprotein phosphatase (PPP) family, the protein phosphatase $Mg^{2+}$- or $Mn^{2+}$-dependent (PPM) family, and the protein tyrosine phosphatase (PTP) (*Barford et al., 1998*). We first examined the effect of pretreatment of the phosphatase inhibitors, which would elevate the basal phosphorylation level of TRPV2 and compromise the subsequent enhancing effect of $Mg^{2+}$ on current responses. As shown in *Figure 5A–B and a* significant impact was observed with PTP inhibitor 1 (2-bromo-4'-hydroxy acetophenone) and PTP inhibitor 2 (4-(bromoacetyl)anisole), but not PPP inhibitors salubrinal, LB-100, cyclosporin A, cantharidin, nor the PPM inhibitor CCT007093. We then confirmed that inhibition of tyrosine dephosphorylation by PTP inhibitors indeed increased tyrosine phosphorylation levels of TRPV2 (*Figure 5C–D*). Besides, we found that in BMDM, the upregulation of tyrosine phosphorylation of TRPV2 caused by PTP inhibitors induced a left-shift of the concentration-response curve to agonist application (*Figure 5E–F*). The corresponding $EC_{50}$ values were 0.18 ± 0.01 and 0.09 ± 0.01 mM in the presence of PTP inhibitor 1 or 2, respectively, compared to $EC_{50}$ = 0.55 ± 0.01 mM under control condition. Conversely, TRPV2(Y335/471/525F) mutant deficit in $Mg^{2+}$ influx showed no significant change in the presence of PTP inhibitors (*Figure 5G*).

We next determined the effect of PTP-mediated dephosphorylation of TRPV2 on its temperature sensitivity. We employed an ultrafast infrared laser system capable of delivering a short temperature pulse surrounding BMDMs. *Figure 5H–P* illustrates heat-activated currents of TRPV2 treated with DMSO (*Figure 5H–J*), PTP inhibitor 1 (*Figure 5K–M*), and PTP inhibitor 2 (*Figure 5N–P*), respectively. Remarkably, we observed that boosting tyrosine phosphorylation lowered the thermal activation threshold of TRPV2 by ~12°C. Similar results were obtained for TRPV2 channels expressed in HEK293T heterologous expression systems (*Figure 5—figure supplement 1*). Taken together, these results support that tyrosine phosphorylation promotes both the chemical and thermal sensitivities of TRPV2, which are both controlled by phosphatase dephosphorylation.

## PTPN1 phosphatase controls tyrosine phosphorylation homeostasis

We further determined the subtypes of PTP phosphatases involved in controlling TRPV2 phosphorylation processes. We observed that knocking down of PTPN1 phosphatase by shRNA increased the tyrosine phosphorylation of TRPV2 (*Figure 6A–B*), which increased its sensitivity to the chemical agonist 2-APB (*Figure 6C*). Conversely, no effect was observed following the inhibition of the expression of PTPN2, PTPN11, PTPN12, PTPN14, PTP4A1, or PTEN (*Figure 6C*). As corroboration, downregulating PTPN1 expression to boost the basal phosphorylation level compromised the enhancing effect of subsequently applied $Mg^{2+}$ on TRPV2 current responses (*Figure 6D–E*).

We then investigated the effect of PTPN1 on heat activation of TRPV2, by applying time-locked temperature jumps. Increasing tyrosine phosphorylation by inhibition of the PTPN1-mediated

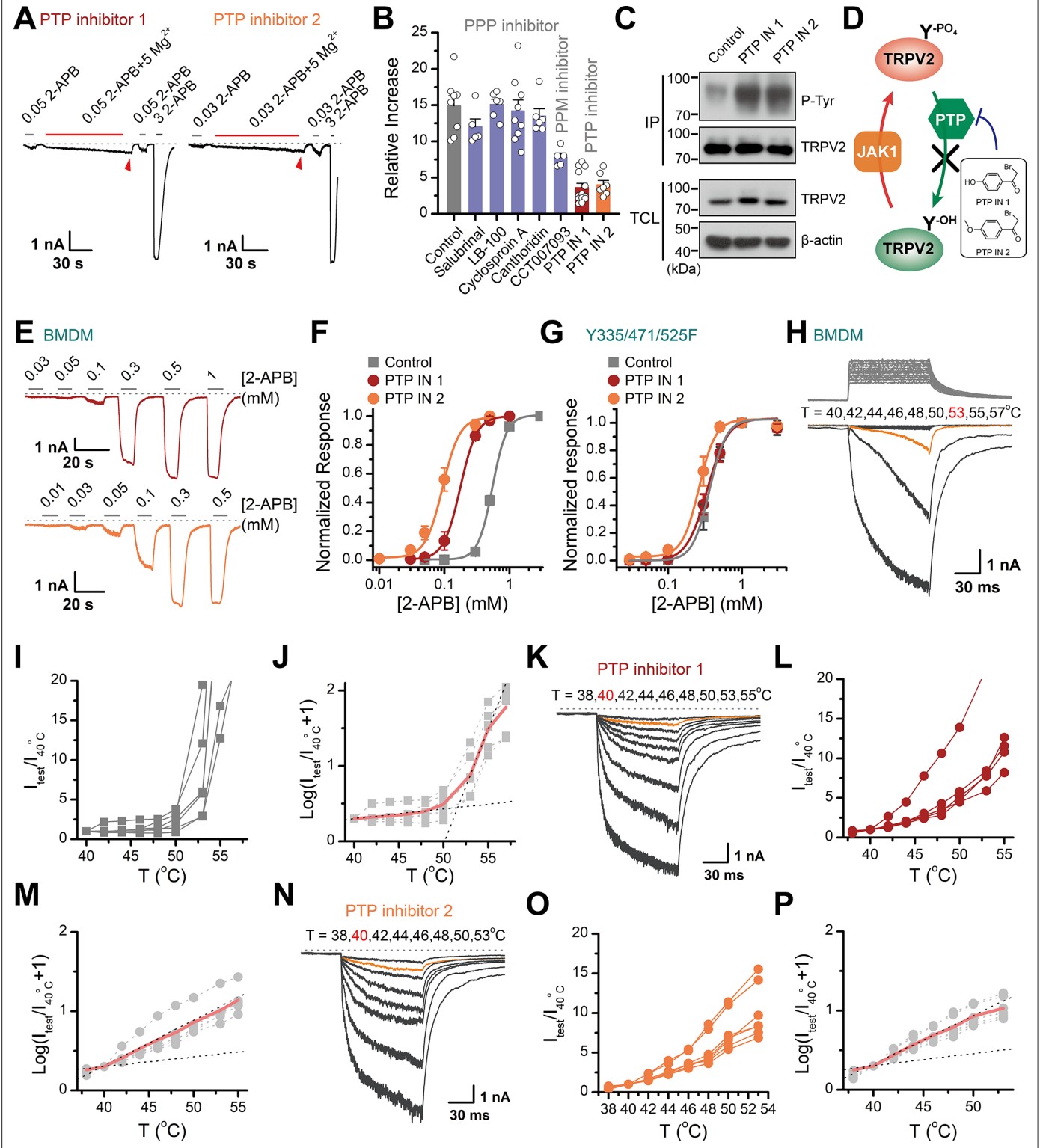

**Figure 5.** Increasing the phosphorylation level of transient receptor potential vanilloid 2 (TRPV2) by inhibition of dephosphorylase activity enhances the channel sensitivity to its stimuli. (**A**) Whole-cell recordings from TRPV2-expressing HEK293T cell were consecutively challenged with 0.3 mM 2-aminoethyl diphenylborinate (2-APB), 0.3 mM 2-APB plus 5 mM $Mg^{2+}$ and 3 mM 2-APB. The cells were pretreated with protein tyrosine phosphatase (PTP) inhibitor 1 and PTP inhibitor 2 for 5 min, respectively. The dotted line indicates zero current level. (**B**) Summary plot of effects of various phosphatase inhibitors on TRPV2 currents. (**C**) Immunoblotting analysis with anti-phosphotyrosine antibody exhibiting tyrosine phosphorylation of immunoprecipitated TRPV2-Flag in HEK293T cells under control conditions and after treatment with PTP inhibitor 1 or PTP inhibitor 2. (**D**) Schematic

*Figure 5 continued on next page*

*Figure 5 continued*

diagram showing increased TRPV2 tyrosine-phosphorylation levels caused by phosphokinase JAK1 or inhibition of PTP activity. (**E**) Representative whole-cell currents evoked by increasing concentrations of 2-APB for rat bone marrow-derived macrophages (rBMDMs). The cells were pretreated with PTP inhibitor 1 (*top*) and PTP inhibitor 2 (*bottom*). (**F**) Dose-response curves of 2-APB. Fitting by Hill's equation resulted in the following: $EC_{50} = 0.55 \pm 0.01$ mM and $n_H = 3.9 \pm 0.2$ for control (n = 6); $EC_{50} = 0.18 \pm 0.01$ mM and $n_H = 3.4 \pm 0.1$ for treatment by PTP inhibitor 1 (n = 6) and $EC_{50} = 0.09 \pm 0.01$ mM and $n_H = 3.3 \pm 0.3$ for treatment by PTP inhibitor 2 (n = 7). (**G**) Concentration-response curves of 2-APB in TRPV2-Y335/471/525F-expressing HEK293T cells under treatment by DMSO, PTP inhibitor 1 or PTP inhibitor 2. Fitting by Hill's equation resulted in the following: $EC_{50} = 0.36 \pm 0.01$ mM and $n_H = 3.8 \pm 0.1$ for control (n = 5); $EC_{50} = 0.34 \pm 0.01$ mM and $n_H = 3.1 \pm 0.1$ for treatment by PTP inhibitor 1 (n = 6) and $EC_{50} = 0.26 \pm 0.01$ mM and $n_H = 3.8 \pm 0.7$ for treatment by PTP inhibitor 2 (n = 6). (**H–J**) Representative current traces, temperature-activation relations, and plot of $\log(I_{test}/I_{40oC}+1)$ determinations for DMSO pretreated rBMDMs. (**K–M**) Representative current traces, temperature-activation relations, and plot of $\log(I_{test}/I_{40oC}+1)$ determinations for PTP inhibitor 1 pretreated rBMDMs. (**N–P**) Representative current traces, temperature-activation relations, and plot of $\log(I_{test}/I_{40oC}+1)$ determinations for PTP inhibitor 2 pretreated rBMDMs.

The online version of this article includes the following source data and figure supplement(s) for figure 5:

**Source data 1.** Uncropped, unedited blots for *Figure 5C*.

**Figure supplement 1.** Inhibition of protein tyrosine phosphatase (PTP) activity by inhibitors enhanced the transient receptor potential vanilloid 2 (TRPV2) sensitivity to 2-aminoethyl diphenylborinate (2-APB) and heat in TRPV2-expressing HEK293T cells.

dephosphorylation significantly decreased the temperature threshold of TRPV2 activation (*Figure 6F–K*). These data suggest that PTPN1 phosphatase restrains basal phosphorylation levels of TRPV2 to regulate its function.

## Discussion

TRPV2 ion channel senses a wide range of sensory inputs and is an essential player in physiopathological contexts. In the present study, we delineate a hitherto unrecognized tyrosine phosphorylation module that defines the homeostatic sensitivity of TRPV2 ion channel (*Figure 6—figure supplement 1*).

Our data show that $Mg^{2+}$ modulates tyrosine phosphorylation levels of the TRPV2 channel protein, and thereby also its activity. This observation mirrors the established role of $Mg^{2+}$ in the regulation of phosphokinase catalytic activities and the regulation of diverse ion channels including NMDA receptors (*Antonov and Johnson, 1999*) and TRP ion channels (*Cao et al., 2014*; *Lee et al., 2005*; *Luo et al., 2012*; *Obukhov and Nowycky, 2005*; *Yang et al., 2014*). $Mg^{2+}$ is the most abundant divalent cation in living cells with the intracellular concentration of 10–30 mM. In the cytosol, the majority of $Mg^{2+}$ is bound to ribosomes, polynucleotides, and ATP, resulting in the free $Mg^{2+}$ concentration of about 0.3–1.2 mM (*de Baaij et al., 2015*; *Funato et al., 2014*; *Moomaw and Maguire, 2008*). $Mg^{2+}$ participates in a wide range of fundamental cellular reactions and its deficiency may lead to many disorders. For instance, it has been reported that magnesium deficiency caused by genetic deficiencies in MAGT1 impairs anti-virus immune response which can be restored by intracellular free magnesium supplementation (*Chaigne-Delalande et al., 2013*). Interestingly, they also found that the concentration of intracellular free $Mg^{2+}$ can be increased by long-term $Mg^{2+}$ supplementation (*Chaigne-Delalande et al., 2013*). As a more efficient way to alter intracellular $Mg^{2+}$ concentrations, $Mg^{2+}$ can permeate into the cell through ion channels such as TRPM6, TRPM7, or/and magnesium transporters like MagT1 (*Deason-Towne et al., 2011*; *Goytain and Quamme, 2005*; *Voets et al., 2004*). Supplying the $Mg^{2+}$-chelator EDTA through patch-clamp glass pipette, our data suggest that transient $Mg^{2+}$ buildup on the intracellular side is required for shifting the tyrosine phosphorylation level. This mechanism differs from the action of $Mg^{2+}$ on TRPV1 channels, where a high concentration of $Mg^{2+}$ potentiates the TRPV1 activity from the extracellular side but inhibits TRPV1 currents from the intracellular side (*Cao et al., 2014*; *Yang et al., 2014*).

In addition, we noticed that the addition of 5 mM $Mg^{2+}$ to the pipette solution was less effective than the effect of $Mg^{2+}$ influx from outside the cell through the channel. One possible reason is that local transient concentration changes in $Mg^{2+}$ are more capable of inducing enzymatic reactions, as local $Ca^{2+}$ sparklets are more likely to facilitate subsequent reactions (*Amberg et al., 2007*). It could also be that a uniform global increase in $Mg^{2+}$ concentration may trigger both phosphorylation and dephosphorylation, which may compromise the degree of TRPV2 phosphorylation.

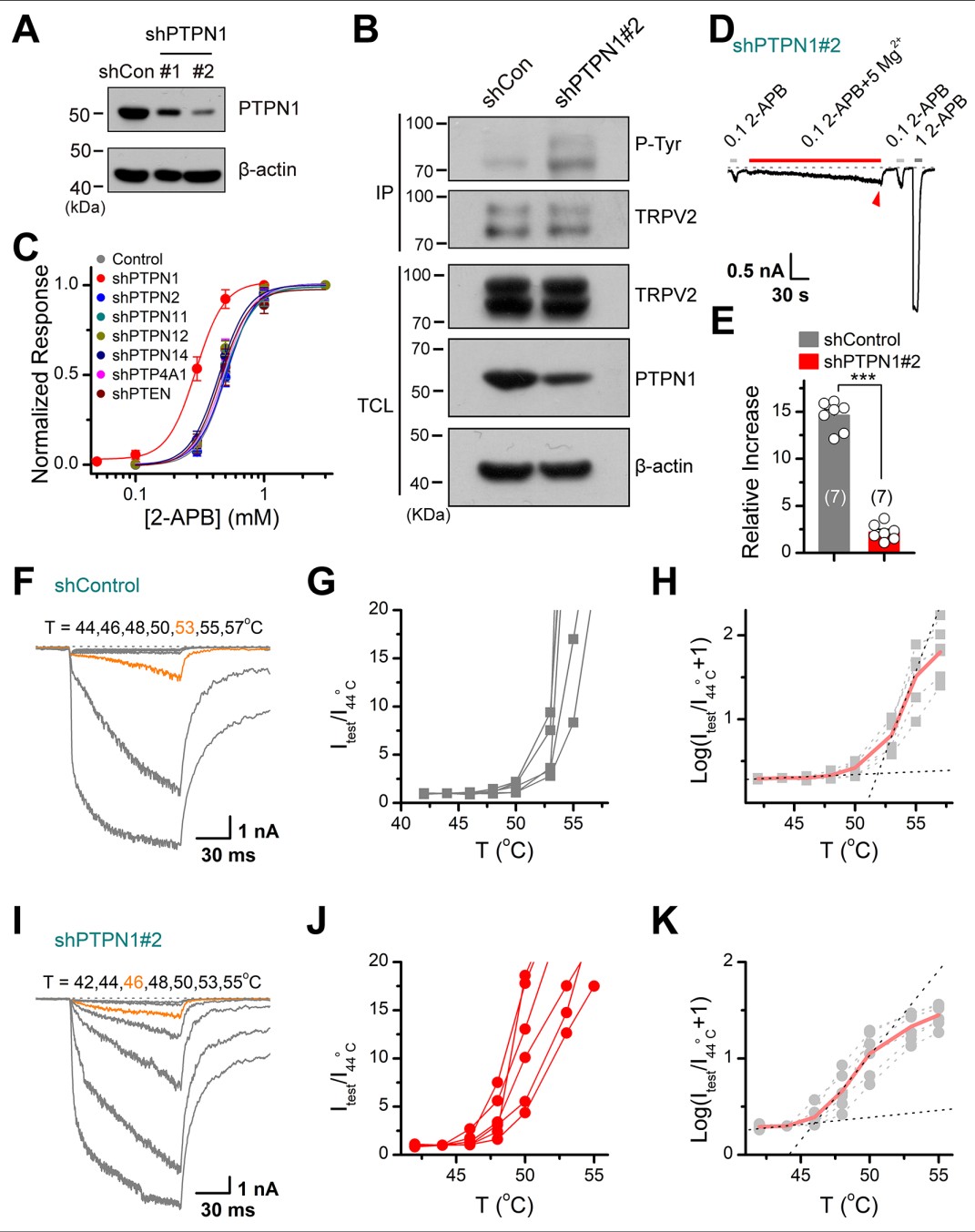

**Figure 6.** Protein tyrosine phosphatase non-receptor type 1 (PTPN1) is a phosphatase that mediates the dephosphorylation of transient receptor potential vanilloid 2 (TRPV2). (**A**) Immunoblot analysis (with anti-PTPN1 or anti-β-action) of HEK293T cells transfected for 48 hr with PTPN1-targeting shRNA (shPTPN1#1 and shPTPN1#2) or shControl to test knockdown efficiency of shRNA. (**B**) Immunoblot analysis of the tyrosine phosphorylation level of TRPV2 in HEK293T cells transfected with shControl or shPTPN1#2 for 48 hr. (**C**) Concentration-response curves of 2-aminoethyl diphenylborinate (2-APB). Whole-cell recordings were performed in HEK293T transfected with various protein tyrosine phosphatase-targeting shRNA. (**D**) Whole-cell recordings in TRPV2-expressing HEK293T cells that were transfected for 48 hr with shPTPN1#2 showing the response to 0.1 mM 2-APB, 0.1 mM 2-APB plus 5 mM $Mg^{2+}$ and 1 mM 2-APB. The dotted line indicates zero current level. (**E**) Comparison of relative changes under different conditions. p = 3.88E-10 by unpaired Student's $t$-test. (**F–H**) Representative current traces, temperature-activation relations, and plot of $\log(I_{test}/I_{44oC}+1)$ determinations for TRPV2-expressing HEK293T cells which were transfected for 48 hr with shControl. (**I–K**) Representative current traces, temperature-activation relations, and plot of $\log(I_{test}/I_{44oC} + 1)$ determinations for TRPV2-expressing HEK293T cells which were transfected for 48 hr with shPTPN1#2.

*Figure 6 continued on next page*

*Figure 6 continued*

The online version of this article includes the following source data and figure supplement(s) for figure 6:

**Source data 1.** Uncropped, unedited blots for *Figure 6A*.

**Source data 2.** Uncropped, unedited blots for *Figure 6B*.

**Figure supplement 1.** Tyrosine phosphorylation sets the agonist and heat sensitivity of transient receptor potential vanilloid 2 (TRPV2).

We reveal that $Mg^{2+}$-mediated enhancing effect on TRPV2 current responses is tuned by JAK1 kinase and PTPN1 phosphatase at Y335, Y471, and Y525 molecular sites. Tyrosine phosphorylation of TRPV2 controls not only its sensitivity to chemical stimulations but also its thermal activation threshold. Temperature sensing is essential to survive and adapt since failure to avoid noxious temperatures can cause fundamental tissue damage. TRPV1, TRPV2, TRPV3, TRPV4, and TRPM2 channels together sense a broad temperature range spanning from physiological warmness to noxious hotness. The physiological role of the TRPV1 channels in thermosensation has been demonstrated by the knockout of the TRPV1 channels in mice (*Garami et al., 2011*). However, the physiological role of the TRPV2 channels remains unclear while it is responsive to noxious heat (>52°C) in heterologous systems. We here demonstrate that enhancing the tyrosine phosphorylation levels of TRPV2 protein lowers its thermal threshold to a near-body temperature level (~40°C). TRPV2 might act as a heat thermo-sensor in physiopathological conditions when encountering either or both $Mg^{2+}$ surges and upregulated tyrosine phosphorylation (*Yu et al., 2011*). For instance, intracellular free $Mg^{2+}$ can be increased by adenosine triphosphate (ATP) depletion induced by either mitochondrial deficits (*Kubota et al., 2003*) or cell reactive states that consume a high amount of cytosolic ATP (*Brocard et al., 1993*; *Gaussin et al., 1997*). In addition to tyrosine phosphorylation, oxidation of methionine residues or other potential endogenous modulators would independently or synergistically modulate TRPV2 channel sensitivity (*Fricke et al., 2019*).

TRPV2 is expressed abundantly in the cells of the immune system, such as macrophages, osteoclasts, mast cells, and neutrophils (*Kojima and Nagasawa, 2014*). It has been proven that TRPV2 has a critical role in immunity response, such as phagocytosis of macrophages and degranulation of mast cells (*Link et al., 2010*; *Zhang et al., 2012*). JAK1 is also abundantly in the immune cells. *JAK1-/-* mice died soon after birth, the development of lymphocytes was severely impaired, and the mature B cells in thymocytes and spleen were significantly reduced, suggesting that JAK1 plays an important role in immune development and process (*Rodig et al., 1998*). Based on these findings, JAK1-TRPV2 axis may have a regulatory effect on immune processes.

Protein post-translational modification represents a main endogenous regulatory mechanism of ion channels and immune signaling, by changing the plasma membrane expression or altering the biophysical properties of the channels. PKA-mediated phosphorylation of the TRPV1 channels and the TRPV2 channels have been proposed (*Jeske et al., 2008*; *Stokes et al., 2004*). Phosphorylation of TRPV1 channels via PKC-related pathway or Src-related pathway was reported to mediate TRPV1 surface expression level (*Studer and McNaughton, 2010*; *Zhang et al., 2005*). Differentially, our data suggest that tyrosine phosphorylation of TRPV2 directly alters its biophysical properties without changing the expression of TRPV2 on the plasma membrane.

By specifically perturbing the JAK1-mediated phosphorylation and PTPN1-mediated dephosphorylation, we could substantially alter the chemical and thermal sensitivity of TRPV2 ion channel. Thus, TRPV2 channel sensitivity is maintained at the homeostatic point by dynamically balanced phosphorylation/dephosphorylation processes. The $Mg^{2+}$-enhanced TRPV2 current responses are quickly reverted (*Figure 1A–F*), suggesting that the endogenous phosphatase activity of PTPN1 is high. As such, TRPV2 is likely maintained at a low level of phosphorylation in basal conditions.

## Materials and methods

### Key resources table

| Reagent type (species) or resource | Designation | Source or reference | Identifiers | Additional information |
|---|---|---|---|---|
| Antibody | Anti-Phosphotyrosine antibody (Rabbit monoclonal) | Abcam | Cat#ab179530; RRID: AB_828379 | WB (1:1000) |

| Reagent type (species) or resource | Designation | Source or reference | Identifiers | Additional information |
|---|---|---|---|---|
| Antibody | Anti-Phospho-(Ser/Thr)Phe antibody (Rabbit polyclonal) | Abcam | Cat#ab17464; RRID: AB_443891 | WB (1:1000) |
| Antibody | Anti-Flag antibody (Rabbit polyclonal) | Proteintech | Cat#20543–1-AP; RRID: AB_11232216 | WB (1:3000) |
| Antibody | Anti-TRPV2 antibody (Rabbit polyclonal) | Alomone Labs | Cat#ACC-032; RRID: AB_2040266 | WB (1:500), IP (1:200) |
| Antibody | Anti-mouse IgG (H+L) (Goat polyclonal) | Jackson Immunoresearch | Cat#115-035-003; RRID: AB_10015289 | (5 µg) |
| Antibody | Anti-rabbit IgG (H+L) (Goat polyclonal) | Jackson Immunoresearch | Cat#111-005-003; RRID: AB_2337913 | (5 µg) |
| Antibody | Anti-JAK1 antibody (Rabbit monoclonal) | Abcam | Cat#ab133666 | WB (1:1000) |
| Antibody | Anti-PTPN1 antibody (Rabbit monoclonal) | Abcam | Cat#ab244207; RRID: AB_2877148 | WB (1:1000) |
| Antibody | Anti-Flag Affinity Gel (Mouse monoclonal) | Bimake | Cat#B23102; RRID: AB_2728745 | (15 µl) |
| Other | *ProteinIso* Protein G Resin | TransGen | Cat#DP401 | (30 µl) |
| Chemical compound, drug | 2-APB | Sigma-Aldrich | Cat#D9754, CAS: 524-95-8 | TRPV2 agonist |
| Chemical compound, drug | $MgCl_2 \cdot 6H_2O$ | Sigma-Aldrich | Cat#M2393, CAS: 7791-18-6 | |
| Chemical compound, drug | $Na_2$-ATP | Sigma-Aldrich | Cat#A2383; CAS: 34369-07-8 | |
| Chemical compound, drug | EDTA | Biosharp | Cat#BS107; CAS: 60-00-4 | |
| Chemical compound, drug | AMP-PNP | Sigma-Aldrich | Cat#A2647; CAS: 25612-73-1 | |
| Chemical compound, drug | MK-2206 | TargetMol | Cat#T1952; CAS: 1032350-13-2 | Akt inhibitor |
| Chemical compound, drug | Staurosporine | TargetMol | Cat#T6680; CAS: 62996-74-1 | PKC inhibitor |
| Chemical compound, drug | KN-93 Phosphate | TargetMol | Cat#T2606; CAS: 1188890-41-6 | CaMKII inhibitor |
| Chemical compound, drug | D4476 | TargetMol | Cat#T2449; CAS: 301836-43-1 | CK1 inhibitor |
| Chemical compound, drug | U0126-EtOH | TargetMol | Cat#T6223; CAS: 1173097-76-1 | MEK1/2 inhibitor |
| Chemical compound, drug | Ruxolitinib | TargetMol | Cat#T1829; CAS: 941678-49-5 | JAK1 inhibitor |
| Chemical compound, drug | Salubrinal | TargetMol | Cat#T3045; CAS: 405060-95-9 | PP1 inhibitor |
| Chemical compound, drug | LB-100 | MCE | Cat#HY-18597; CAS: 1632032-53-1 | PP2A inhibitor |
| Chemical compound, drug | Cyclosproin A | TargetMol | Cat#T0945; CAS: 59865-13-3 | PP2B inhibitor |
| Chemical compound, drug | Cantharidin | Aladdin | Cat#c111020; CAS: 56-25-7 | PP1 and PP2A inhibitors |
| Chemical compound, drug | CCT007093 | TargetMol | Cat#T1927; CAS:176957-55-4 | PPM1D inhibitor |
| Chemical compound, drug | PTP inhibitor 1 | TargetMol | Cat#T7084; CAS: 2491-38-5 | PTPs inhibitor |
| Chemical compound, drug | PTP inhibitor 2 | TargetMol | Cat#T7541; CAS: 2632-13-5 | PTPs inhibitor |
| Cell lines (species) | Human embryo kidney (HEK) 293T (human) | ATCC | Cat#CRL-3216; RRID: CVCL_0063 | |
| Software, algorithms | QStudio | Developed by Dr Feng Qin from University of New York at Buffalo | | |

| Reagent type (species) or resource | Designation | Source or reference | Identifiers | Additional information |
|---|---|---|---|---|
| Software, algorithms | Micro-Manager 1.4 | Vale Lab, UCSF | | |
| Software, algorithms | Clampfit | Molecular Devices, Sunnyvale, CA | | |
| Software, algorithms | IGOR | Wavemetrics, Lake Oswego, OR | | |
| Software, algorithms | SigmaPlot | SPSS Science, Chicago, IL | | |
| Software, algorithms | OriginPro | OriginLab Corporation, Northampton, MA | | |
| Software, algorithms | ImageJ | *Schneider et al., 2012* | | |

## Cell lines

HEK293T cell line used in this study was from the American Type Culture Collection and Thermo Fisher, authenticated by STR locus and tested negative for mycoplasma contamination. HEK293T cells were grown in Dulbecco's modified Eagle's medium (DMEM, Thermo Fisher Scientific, Waltham, MA) containing 4.5 mg/ml glucose, 10% heat-inactivated fetal bovine serum (FBS), 1% penicillin-streptomycin, and were incubated at 37°C in a 5% $CO_2$ humidified incubator. Cells grown into ~80% confluence were transfected with the desired DNA constructs using Lipofectamine 2000 (Invitrogen, Carlsbad, CA) following the protocol provided by the manufacturer. Transfected cells were reseeded on poly-L-lysine-coated glass coverslips for electrophysiological experiments. Experiments took place usually 12–24 hr after transfection.

## cDNA constructs and mutagenesis

WT rat TRPV2 (rTRPV2) was generously provided by Dr Feng Qin (State University of New York at Buffalo, Buffalo, NY). JAK1 was a gift from Dr Hongbing Shu (Medical Research Institute, Wuhan University). All mutations were generated using the overlap-extension polymerase chain reaction method as previously described (*Wang et al., 2020*) and were verified by DNA sequencing. Oligo DNAs targeting JAK1, PTPN1, and several PTPs were synthesized, annealed, and inserted into pLKO.1 vector. The sequences of JAK1 shRNA are as follows: for rat JAK1 shRNA: #1, 5'-GCCCTGAGTTAC TTGGAAGAT-3'; #2, 5'-CGGTCCAATC TGCACAGAATA-3'; #3, 5'-GCAGAAACCAAATGTTCTTCC-3'; for human JAK1 shRNA: #1, 5'-GAGACTTCCATGTTACTGATT-3'; #2, 5'-GACAGTCACAAGAC TTGTG AA-3'; #3, 5'-GCCTTAAGGAATATCTTCCAA-3'. The sequences of PTPN1 shRNA are as follows: for human PTPN1 shRNA: #1, 5'-TGCGACAGCTAGAATTGGAAA-3'; #2, 5'-GCTGCTCTGCTATATGCCTT A-3'. The sequences of rat TRPV2 shRNA are as follows: #1, 5'-GCATGCTCTGGTAATG ATTGC-3'; #2, 5'-GCTGTTCAAGTTCACCATTGG-3'; #3, 5'-GGAAATCTCCA ACCACCAAGG-3'; #4, 5'-GGAAGTTG CAGAAAGCCATCT-3'.

## Rat and mouse bone marrow-derived macrophages

Bone marrow-derived cells were isolated from 4- to 8-week-old Sprague-Dawley (SD) rats as described (*Zhang et al., 2020*). After the rats were euthanized, the femurs and tibias were collected. The cells were resuspended in bone marrow differentiation media, RPMI1640 supplemented with 1% penicillin-streptomycin, 10% FBS, and 30% L929 cells conditioned medium containing macrophage colony stimulating factor (M-CSF) for 4–6 days to obtain BMDMs. Cells were cultured at 37°C in a classic $CO_2$ incubator with 5% $CO_2$.

All animals were housed in the specific pathogen-free animal facility at Wuhan University and all animal experiments were following protocols approved by the Institutional Animal Care and Use Committee of Wuhan University (No. WDSKY0201804) and adhered to the Chinese National Laboratory Animal-Guideline for Ethical Review of Animal Welfare. The animals were euthanatized with $CO_2$ followed by various studies.

## Preparation of DRG neurons

DRG neurons were prepared for electrophysiological experiments by minor modification of a previously described method (*Tian et al., 2019*). Briefly, 4- to 6-week-old adult SD male rats were deeply anesthetized and decapitated. DRGs together with dorsal-ventral roots and attached spinal nerves were isolated from thoracic and lumbar segments of spinal cords. After removal of the attached nerves

and surrounding connective tissues, DRG neurons were rinsed with ice-cold phosphate buffer saline (PBS). Ganglia were dissociated by enzymatic treatment with collagenase type IA (1 mg/ml), trypsin (0.4 mg/ml), and DNase I (0.1 mg/ml) and incubated at 37°C for 30 min. Then cells were dispersed by gentle titration, collected by centrifuge, seeded onto 0.1 mg/ml poly-L-lysine-coated coverslips, and maintained in DMEM/F12 medium containing 10% FBS, 1% penicillin, and streptomycin. Electrophysiology recordings were carried out ~2–4 hr after plating.

## Electrophysiology

The patch-clamp recording of channel currents was made in either whole-cell or inside-out configuration. Currents were amplified using an Axopatch 200B amplifier (Molecular Devices, Sunnyvale, CA) through a BNC-2090/MIO acquisition system (National Instruments, Austin, TX). Data acquisition was controlled by QStudio developed by Dr Feng Qin at State University of New York at Buffalo. Data were typically sampled at 5 kHz and low-pass filtered at 1 kHz. Recording pipettes were pulled from borosilicate glass capillaries (World Precision Instruments [WPI]) to 2–4 MΩ when filled with 150 mM NaCl solution. The compensation of pipette series resistance (>80%) and capacitance was taken by using the built-in circuitry of the amplifier, and the liquid junction potential between the pipette and bath solutions was zeroed prior to seal formation. All voltages were defined as membrane potentials with respect to extracellular solutions. For whole-cell recording, the bath solution contained the following (in mM): 140 NaCl, 5 KCl, 3 EGTA, 10 HEPES (the pH was adjusted to 7.4 with NaOH). In one set of experiments, the salt of $YCl_2$ (Y means $Mg^{2+}$, $Mn^{2+}$, $Ca^{2+}$, $Ba^{2+}$, $Zn^{2+}$, $Cu^{2+}$, $Ni^{2+}$, $Cd^{2+}$, or $Co^{2+}$) was individually dissolved in deionized water to make stock solutions and subsequently diluted into a basic solution ([in mM] 140 NaCl, 5 KCl, and 10 HEPES, pH 7.4) to make a desired final concentration. The solution containing 10–100 mM $Mg^{2+}$ was prepared from 140 mM NaCl-containing solution by replacing the appropriate NaCl with $MgCl_2$. The internal pipette solution consisted of (in mM): 140 CsCl, 10 HEPES, and 1 ATP-$Na_2$, pH 7.4 (adjusted with CsOH). For inside-out recordings, the bath and pipette solutions were symmetrical and contained (in mM): 140 NaCl, 5 KCl, 10 HEPES, pH 7.4 adjusted with NaOH. For cation substitution experiment, pipette solution contains (in mM): 140 NaCl and 10 HEPES, pH 7.4 adjusted with NaOH. After the whole-cell configuration was obtained, bath solution was replaced with specific cationic solution, and a voltage step pulse was used to measure the reversal potential. Channel activators were diluted into the recording solution at the desired final concentrations and applied to the cell of interest through a gravity-driven local perfusion system. Unless otherwise stated, all chemicals were purchased from Sigma (Sigma, St Louis, MO). Water-insoluble reagents were dissolved in either 100% ethanol or DMSO to make stock solutions and were diluted in the recording solutions at appropriate concentrations before experiments. The final concentrations of ethanol or DMSO did not exceed 0.3%, which did not affect the currents. All experiments except those for heat activation were sampled at room temperature (22–24°C).

## Temperature jump

Fast-temperature jumps were produced by a single emitter infrared laser diode (1470 nm) as previously described (*Yao et al., 2009*). Briefly, the laser diode was driven by a pulsed quasi-CW current power supply (Stone Laser, Beijing, China), and the pulsing of the controller was controlled from a computer through the data acquisition card using QStudio software. Constant temperature steps were generated by irradiating the tip of an open pipette filled with the pipette solution and the current of the electrode was used as a readout for feedback control. The sequence of the modulation pulses was stored and subsequently played back to apply temperature jumps to the cell of interest. The temperature was calibrated offline from the pipette current based on the temperature dependence of electrolyte conductivity. The threshold temperature for heat activation of TRPV2 was determined by the methods as previously described (*Zhang et al., 2018*).

## Immunoprecipitation and Western blot

In brief, cells were collected and lysed in Nonidet P-40 lysis buffer containing 150 mM NaCl, 1 mM EDTA, 1% Nonidet P-40, 1% protease inhibitor cocktail, and 1% phosphatase inhibitor cocktail if needed after washing with PBS. The anti-Flag affinity gel or the appropriate antibodies were added into the lysates and incubated at 4°C for 4 hr or overnight with slow rotation. After being washed three times with prelysis buffer containing 500 mM NaCl, the precipitants were resuspended into 2×

SDS sample buffer, boiled, and subjected to SDS-polyacrylamide gel electrophoresis (SDS-PAGE). Immunoblot analysis was performed with the appropriate antibodies.

## Mass spectrometry analysis

To identify in vivo tyrosine phosphorylation sites of TRPV2, HEK293T cells were transfected with Flag-tagged TRPV2. After 24 hr, the cells were harvested following the treatment with 0.3 mM 2-APB or the combination of 0.3 mM 2-APB and 5 mM $Mg^{2+}$ lasting for 5 min. Flag-TRPV2 was immunoprecipitated by anti-Flag affinity gel and subjected to SDS-PAGE.

The samples were digested with trypsin, enriched by titanium dioxide, and then analyzed by liquid chromatography–tandem mass spectrometry (LC-MS/MS) using a Q Exactive-HF mass spectrometer (Thermo Fisher Scientific). Dynamic modification included oxidation (Met) and phosphorylation (STY), and static modification included carbamidomethylation (Cys). The LC-MS/MS data were processed using Proteome Discoverer (version 2.1, Thermo Fisher Scientific) and searched against the Swiss-prot *Homo sapiens* protein sequence database.

## In vitro kinase assay

In vitro kinase assay was performed as previously described (*Li et al., 2019*). In brief, HEK293T cells were transfected with plasmids encoding Flag-JAK1, Flag-JAK1(K908A), respectively. Cells were lysed with NP-40 lysis buffer and the cell lysates were immunoprecipitated with anti-Flag agarose (Sigma, St Louis, MO). His-tagged TRPV2 and His-tagged TRPV2 (Y335F) were purified from bacteria (*E. coli*) using Ni-Agarose Resin. For the JAK1 in vitro kinase assay in *Figure 3*, Flag-JAK1 was respectively incubated with His-TRPV2 in the kinase buffer (6.25 mM Tris-HCl [pH 7.5], 0.125 mM $Na_3VO_4$, 2.5 mM $MgCl_2$, 0.125 mM EGTA, 0.625 mM DTT, and 0.01% Triton X-100) in the presence of 10 µCi [$^{32}$P]-γ-ATP (Perkin Elmer Company) with a final volume of 20 µl. For the JAK1 in vitro kinase assay in *Figure 4*, His-TRPV2 and His-TRPV2 (Y335F) were incubated with or without Flag-JAK1 and Flag-JAK1(K908A) in the kinase buffer in the presence of 10 µCi [$^{32}$P]-γ-ATP with a final volume of 20 µl. The mixture was incubated at 30°C on a shaker with 300 rpm shaking for 60 min. The reaction mixtures were resolved by SDS-PAGE, and $^{32}$P-labeled proteins were analyzed by autoradiography.

## Assessment of phagocytosis

For phagocytosis assays, BMDMs were incubated with RPMI 1640 medium addition of GFP *E. coli* together with 0.1 or 0.05 mM SKF96365, or 2, 5, and 10 µM Ruxolitinib in six-well translucent plates (JET Biofil, China) for 2 hr at 37°C. After washing two to three times by PBS, the BMDMs were harvested by cell Scrapers, resuspended into PBS, and analyzed by flow cytometry using a CytoFLEX Flow Cytometer (Beckman Coulter, Brea, CA).

## Statistical analysis

Electrophysiological data were analyzed offline with Clampfit (Molecular Devices, Sunnyvale, CA), IGOR (Wavemetrics, Lake Oswego, OR), SigmaPlot (SPSS Science, Chicago, IL), and OriginPro (OriginLab Corporation, Northampton, MA). For concentration-dependent analysis, the modified Hill's equation was used: $Y = A1 + (A2 - A1)/[1+10^{\wedge}(logEC_{50} - X)*n_H]$, in which $EC_{50}$ is the half-maximal effective concentration, and $n_H$ is the Hill's coefficient. All data are expressed as either mean ± standard error of the mean (SEM) or mean ± standard deviation (SD) as stated, from a population of cells (n). Statistical tests of significance were carried out by Student's *t*-test for one-group comparison and two-group comparison or one-way analysis of variance (ANOVA) tests for multiple group comparisons, and $p < 0.05$ was considered statistically significant (*$p < 0.05$, **$p < 0.01$, ***$p < 0.001$).

## Acknowledgements

We are grateful to Drs Xiaolu Zhao, Zan Huang, Yan Wang, and members of Yao lab for critical comments and helpful discussions. We also would like to thank the core facilities of College of Life Sciences at Wuhan University for technical help. This work was supported by grants from the National Natural Science Foundation of China (32171147, 31830031, 31929003, 31871174, and 31671209), and the Fundamental Research Funds for the Central Universities (2042021KF0218).

## Additional information

### Funding

| Funder | Grant reference number | Author |
|---|---|---|
| National Natural Science Foundation of China | 32171147 | Jing Yao |
| National Natural Science Foundation of China | 31830031 | Jing Yao |
| National Natural Science Foundation of China | 31929003 | Jing Yao |
| National Natural Science Foundation of China | 31871174 | Jing Yao |
| National Natural Science Foundation of China | 31671209 | Jing Yao |
| Fundamental Research Funds for the Central Universities | 2042021KF0218 | Jing Yao |
| Wuhan University | | Jing Yao |

The funders had no role in study design, data collection and interpretation, or the decision to submit the work for publication.

### Author contributions

Xiaoyi Mo, Conceptualization, Data curation, Formal analysis, Investigation, Validation, Writing – original draft; Peiyuan Pang, Yulin Wang, Dexiang Jiang, Conceptualization, Data curation, Formal analysis, Investigation, Validation; Mengyu Zhang, Yang Li, Peiyu Wang, Data curation, Formal analysis, Investigation, Validation; Qizhi Geng, Data curation, Investigation, Validation; Chang Xie, Conceptualization, Data curation, Formal analysis, Investigation, Methodology, Validation; Hai-Ning Du, Bo Zhong, Dongdong Li, Conceptualization, Methodology, Validation, Visualization; Jing Yao, Conceptualization, Data curation, Formal analysis, Funding acquisition, Investigation, Methodology, Project administration, Resources, Software, Supervision, Validation, Visualization, Writing – original draft, Writing – review and editing

### Author ORCIDs

Xiaoyi Mo http://orcid.org/0000-0003-0903-1877
Peiyuan Pang http://orcid.org/0000-0001-9609-7473
Dongdong Li http://orcid.org/0000-0002-6731-4771
Jing Yao http://orcid.org/0000-0003-1844-3988

### Ethics

All animals were housed in the specific pathogen-free animal facility at Wuhan University and all animal experiments were following protocols approved by the Institutional Animal Care and Use Committee of Wuhan University (NO. WDSKY0201804) and adhered to the Chinese National Laboratory Animal-Guideline for Ethical Review of Animal Welfare. The animals were euthanatized with $CO_2$ followed by various studies.

### Decision letter and Author response

Decision letter https://doi.org/10.7554/eLife.78301.sa1
Author response https://doi.org/10.7554/eLife.78301.sa2

## Additional files

### Supplementary files
• Transparent reporting form

## Data availability

All major datasets supporting the conclusions of this article has been deposited at Dryad, https://doi.org/10.5061/dryad.41ns1rng6.

The following dataset was generated:

| Author(s) | Year | Dataset title | Dataset URL | Database and Identifier |
|---|---|---|---|---|
| Yao J, Mo X, Pang P, Wang Y, Jiang D, Zhang M, Li Y, Wang P, Geng Q, Xie C, Du H, Zhong B Li D | 2022 | Tyrosine phosphorylation tunes chemical and thermal sensitivity of TRPV2 ion channel | https://dx.doi.org/10.5061/dryad.41ns1rng6 | Dryad Digital Repository, 10.5061/dryad.41ns1rng6 |

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
