## [Editor Report]

This important study by Xiaoyi Mo and collaborators identifies and carefully describes a novel mechanism controlling TRPV2 channel sensitivity to heat and 2-APB through phosphorylation/de-phosphorylation of tyrosine residues. The authors identify the specific kinase and phosphatase involved in this mechanism, as well as the specific residues in the channel whose phosphorylation results in channel sensitization to agonists. Further, evidence is provided that this mechanism is physiologically relevant in bone marrow-derived macrophages from rodents.

---

## [Decision Letter]

**Decision letter after peer review:**

Thank you for submitting your article "Tyrosine phosphorylation tunes chemical and thermal sensitivity of TRPV2 ion channel" for consideration by *eLife*. Your article has been reviewed by 3 peer reviewers, one of whom is a member of our Board of Reviewing Editors, and the evaluation has been overseen by Richard Aldrich as the Senior Editor. The following individual involved in the review of your submission has agreed to reveal their identity: Eric N Senning (Reviewer #2).

The reviewers have discussed their reviews with one another, and the Reviewing Editor has drafted this to help you prepare a revised submission. All three reviewers agreed that the findings in the manuscript are important and novel and that most key conclusions are supported by high-quality experimental data. The reviewers also raised a series of concerns that need to be addressed before publication, which are listed below.

Essential revisions:

Revisions requiring additional experimental work:

1) The authors should include experiments at least in BMDM and TRPV2-expressing HEK293 cells in which 0.3 mM 2-APB is applied for the same duration as 0.3mM 2APB + magnesium. Ideally, this should have been done in all experiments, but I realize this would take a significant amount of effort and that given the rest of the evidence provided, it seems unlikely that this will affect the conclusions.

2) In figure 2 the authors demonstrate that Mg applied alone does not activate TRPV2 in the inside-out configuration (Figure 2G-H). The authors should include an inside-out experiment to address whether intracellularly applied 2-APB together with magnesium have a sensitizing effect.

3) Ion permeability of TRPV2 is not very well explored in previous reports. Thus, permeation of Mg through TRPV2 should be convincingly demonstrated by performing measurements of the reversal potential in different concentrations of extracellular magnesium.

Other essential revisions, including experiments that are suggested, but not required for publication:

4) The authors included ATP in the pipette in most of their experiments, which could be expected to be essential for observing the sensitizing effect of magnesium. This needs to be discussed. It would strengthen the manuscript if the authors included experimental data to directly address whether ATP in the pipette is required for observing the sensitizing effect of magnesium co-applied with 2-APB.

5) The manuscript would benefit from additional editing to improve readability. A more concisely worded introduction that begins with the general properties of TRPV2 and then highlights our gaps in understanding, especially in any immunological role, would improve the transition into the summarized findings. The underlying hypothesis leading the authors to challenge TRPV2 with high concentrations of extracellular Mg should be (better) stated. There are many ways to explore the role of protein phosphorylation in the activity of ion channels, the authors applied many elegant techniques in this study. The starting point with extracellular Mg is somewhat odd, giving the slight impression that the authors did not have protein phosphorylation in mind when they started this study. The discussion should review and discuss the results in a manner consistent with the order of the figures. I would recommend discussing the Mg^2+^ dependent TRPV2 results first and the phosphorylation effect next.

6) In general, the wording in figure descriptions needs to be more concise and clearly relatable to figure legends. For example, Figure 1B indicates the response to washout but the figure description indicates responses to 0.3mM 2-APB with either 0 or 5 mM Mg^2+^. This discrepancy needs to be clarified. I would recommend small colored triangles (blue, red) in the current trace of Figure 1A, which indicate the time points at which the summary data measurement is obtained. This recommendation should be extended to all examples of summary plots that employ the same type of comparison (ie. Figure 1D, F, etc.).

7) The authors should clarify the duration of the exposure to 2APB and magnesium in all experiments where this could be an important factor affecting the results (see public review). The duration of stimulation should be the same or very similar at least for all key experiments – Figure 3E, Figure 3J and K, Figure 4A, G, H, and I, Figure 5B.

8) All figures where current vs temperature relations at different conditions are compared should be normalized at the same temperature for all data, and the threshold calculated accordingly. Examples of this can be found in Zhang et al. (PNAS, 2017; https://www.pnas.org/doi/suppl/10.1073/pnas.1717192115).

9) Line 15-16 on page 5. The authors state that the Mg-induced effect could not be washed out. Looking at the figures, however, it becomes clear that the washout period was only about 10s or so. This is much too short to say anything about if the Mg-induced sensitization of TRPV2 is reversible or not. This needs to be acknowledged and discussed.

10) On page 6, lines 9-12 the authors state that Mg is the only cation exhibiting a "more profound" effect on TRPV2. When looking at Figure 1 supplement 2 however, it appears other cations also have interesting effects on channel activity. The authors should comment on this.

11) The authors imply that increased intracellular Mg-concentration accounts for channel phosphorylation. Why did the authors not perform experiments with different – pre-set – concentrations of Mg in the pipette solution? These experiments would be straightforward to perform, and they would make the approach with the TRPV2-E609/E614 mutant less critical.

12) The authors should remove the calcium imaging data and instead provide measurements of the reversal potential to show whether the mutations effectively reduce permeability to magnesium. Alternatively, all data for the E609/614Q should be removed from the manuscript.

13) The authors should explain the smear that appears on the third lane in the in vitro kinase assay (Figure 3F).

14) The authors should provide additional information on the mass-spectrometry results to highlight the relevant observations that support phosphorylation at Y335.

15) The authors should remove the phagocytosis assay from the manuscript, or alternatively, provide additional data with siRNA-dependent knockdown of TRPV2 channel expression to show specificity. SKF96365 is non-specific and more often employed to inhibit TRPC-channels. It may also block TRPV2. 100 µM SKF is a very high concentration that may even be cytotoxic. It would strengthen the manuscript if the authors provided whole-cell patch-clamp recordings of BMDM treated with TRPV2 siRNA to show that 2-APB responses are either absent or significantly reduced. Although the consistency between BMDM and HEK293 cell experiments gives confidence in the authors' conclusions, it should be considered that 2-APB is a highly non-specific agonist and that primary macrophages express other ion channels in addition to TRPV2.

16) The same concerns about the specificity of the findings apply to the current recordings with DRG neurons and BMDM – 2-APB is a non-selective agonist of many ion channel types, including TRP channels also expressed in DRG neurons. Ideally, the authors should have included DRG and BMDM recordings from TRPV2 KO animals, or from cells treated with siRNA. In the absence of this data, the authors should at least discuss these shortcomings.

17) The authors should provide additional discussion on the possible physiological implications of their findings. Importantly, the authors should discuss whether the magnesium dependence that they observe is consistent with physiological magnesium concentrations in the cytosol, and whether there are any known signaling pathways that could trigger the activity of JAK1 or PTPN1. The discussion about any Mg^2+^ dependent process as physiological seems overly speculative without additional experiments that maintain physiological conditions throughout experiments. It would be better to highlight the Mg^2+^ effect as an excellent tool to explore downstream modulation by cell signaling events. Any speculative aspect of the Mg^2+^ effect that is physiological should be clearly labeled as such and limited in scope.

18) The authors state that three residues (Y335, Y471, Y525) are conserved in other mammalian species. Human TRPV2 is known to often display clear species-specific differences as compared to rodent TRPV2. It would strengthen the manuscript if the authors provided data to address whether human TRPV2 channels are also subject to the regulatory mechanism described in the manuscript. However, this is not required and we leave it up to the authors to decide. Similarly, it would also be interesting if the authors examined whether this effect is conserved in TRPV1, TRPV3 and TRPV4 channels, but we realize this might be out of scope for the present manuscript.

19) The 2-APB concentrations in the figure and the figure legend are not consistent in Figure5A.

Figures 5A/6D: Explain why the concentration change to 0.03 mM/ 0.1 mM 2APB are necessary compared to earlier experiments (0.3 mM) and provide examples of a minimally effective phosphatase such as LB-100/ and shControl as representative traces.

20) Page 2, line 3: it is unclear what the peripheral or central functions are. TRPV2 plays important roles in many non-neuronal tissues.

21) The role of TRPV2 in thermo-, mechano- and osmo-sensing (page 3, lines 6-7 and 11-12) is controversial. The authors should acknowledge this in their introduction.

22) Page 4, lines 20-21: I suggest removing the statement about therapeutics. Alternatively, the authors should be more specific about what pathology they are referring to.

23) Page 18, lines 1-2: I do not think that any of the findings in the manuscript indicate that long-term supplementation with magnesium can increase its cytosolic concentration.

[Editors' note: further revisions were suggested prior to acceptance, as described below.]

Thank you for resubmitting your work entitled "Tyrosine phosphorylation tunes chemical and thermal sensitivity of TRPV2 ion channel" for further consideration by *eLife*. Your revised article has been evaluated by Richard Aldrich (Senior Editor), a Reviewing Editor, and one of the original reviewers.

We congratulate you on this important study, which has become significantly improved after the first revision. Several additional experiments have been included, which together with modifications to the text and figures, address most of the concerns raised in the first review. Some points remain to be fully addressed before publication and are outlined below. None of these points involve additional experimental work.

Essential revisions:

1) Current vs temperature relations in the manuscript figures are normalized differently between the channel constructs that are being compared. The threshold for heat activation appears to be calculated based on this same normalization scheme, which is different from what is indicated in the Methods section ("Temperature Jump": line 536), done in accordance with the technique of Zhang et al. (2018). We appreciate that the example in R3 showed close agreement between their previous method and the preferred one that normalizes to a common temperature point. However, if their methods section claims the use of the optimal approach, then this should be applied consistently in all figures displaying current-temperature relations.

2) Please include zero-current lines for all current traces, so that readers can assess the magnitude of the leak currents in each recording.

3) The new reversal potential measurements with external Na^+^ and Mg^2+^ show that whereas the E609/E614Q double mutant channel has lower Mg^2+^-permeability than WT, Mg^2+^ can still permeate through the mutant channel at least to the same extent as Na^+^ does. This seems difficult to reconcile with the strong negative effect that the double mutant has on the potentiation caused by external Mg^2+^ in Figure 2C and D. Two alternative explanations for the observed results that were not tested are that the E609/E614Q mutations prevent phosphorylation of key tyrosine residues, or that channels lose to capacity to become sensitized by tyrosine phosphorylation. Because these data can be confounding unless additional experimental work is included, we ask the authors to remove these data from the manuscript, or tone down the significance of these findings and mention alternative explanations for the observations that could not be ruled out by the data.

4) The data in Figures R1 (exposure of inside-out patches to Mg^2+^ and 2-APB applied together) and R2 (effect of ATP in the pipette) should be included in the manuscript as a supplement. These results add important information that is not conveyed elsewhere. The data in Figure R5 should also be included in the manuscript as a Supplementary Figure because it highlights an important methodological constraint, namely that the inclusion of Mg^2+^ in the pipette is much less effective in sensitizing TRPV2 channels than when applied from the extracellular side.

5) The apparent affinity for 2-APB seems to be increased in TRPV2 channels containing the triple mutation Y335/471/525F. This is contrary to what is expected from a mutation that mimics a non-phosphorylated state and runs counter to the main conclusions of the manuscript. This apparent discrepancy should be discussed.

6) The text contains several typos and stylistic errors that should be fixed. Here are some that I found, with suggested edits already incorporated:

– "whether TRPV2 can function as a temperature sensor or a mechanical sensor in physiology still remains in debate."

– "In mast cells, TRPV2-mediated calcium flux stimulates protein kinase A (PKA)-dependent proinflammatory degranulation (Stokes 68 et al., 2004)."

– "Here we show that the regulator of phosphokinases magnesium (Mg^2+^), exerts an enhancing effect on both the chemical and thermal sensitivity of TRPV2 endogenously expressed in rat bone marrow-derived macrophages."

– "Considering that TRPV2 is abundantly and functionally expressed in macrophages where other types of TRPV channels are barely detectable (Figure 1 —figure supplement 1) (Link et al., 103 2010; Nagasawa et al., 2007), we used rat bone marrow-derived macrophages (rBMDMs) as an endogenous cell system to record TRPV2 currents."

– "Our data show that Mg^2+^ modulates tyrosine phosphorylation levels of the TRPV2 channel protein, and thereby also its activity."

– "For instance, it has been reported that magnesium deficiency caused by genetic deficiencies in MAGT1 impairs anti-virus immune response".

– "the cell through ion channels such as TRPM6, TRPM7, or/and magnesium transporters like MagT1".

– "Using a TRPV2 mutant deficient in Mg^2+^ permeation and supplying the Mg^2+^-chelator EDTA through the patch-clamp glass pipette, our data suggest that transient Mg".

*Reviewer #1 (Recommendations for the authors):*

I congratulate the authors for the outstanding job of addressing concerns raised in the review and incorporating the reviewers' suggestions in the revised version of their work. Several additional experiments have been included, and the text has been revised and made much clearer, which together has significantly improved the manuscript. Whereas all my major concerns have been addressed, I still have a few minor points that are relevant and can further improve the work:

1) Please include zero-current lines for all current traces, so that readers can assess the magnitude of the leak currents in each recording.

2) The new reversal potential measurements with external Na^+^ and Mg^2+^ show that whereas the E609/E614Q double mutant channel has lower Mg^2+^-permeability than WT, Mg^2+^ can still permeate through the mutant channel at least to the same extent as Na^+^ does. This seems difficult to reconcile with the strong negative effect that the double mutant has on the potentiation caused by external Mg^2+^ in Figure 2C and D. It is possible that the E609/E614Q mutations also disrupt the sensitizing effect of phosphorylation on channel activity, a possibility that was not tested. An experiment should be included to test whether E609/E614Q channels are still sensitized by Tyr phosphorylation, which could be done in inside-out patches exposed cytosolic Mg^2+^ and 2-APB, although other alternative experiments would be possible. Alternatively, the data pertaining to E609/E614Q channels can be removed from the manuscript without affecting the main conclusions.

3) The data in Figure R1 (exposure of inside-out patches to both Mg^2+^ and 2-APB) should be included in the manuscript as a supplement, or at least the results mentioned together with a 'data not shown' statement. These results add important information that is otherwise not conveyed by any of the other data included. The observations on Figure R2 should be referred to in the manuscript at least as 'data not shown'. The data on Figure R5 should be included in the manuscript, as it highlights an important methodological point that would be useful for other scientists when trying to reproduce the results shown here, namely that the inclusion of Mg^2+^ in the pipette is much less effective in sensitizing TRPV2 channels than when the cation is applied from the extracellular side.

4) The apparent affinity for 2-APB seems to be increased in TRPV2 channels containing the triple mutation Y335/471/525F. This is contrary to what is expected from a mutation that mimics a non-phosphorylated state and runs counter to the main conclusions of the manuscript. This apparent discrepancy must be discussed.

5) the text contains several typos and stylistic errors that should be fixed. Here are some that I found, with suggested edits already incorporated:

– "whether TRPV2 can function as a temperature sensor or a mechanical sensor in physiology still remains in debate."

– "In mast cells, TRPV2-mediated calcium flux stimulates protein kinase A (PKA)-dependent proinflammatory degranulation (Stokes 68 et al., 2004)."

– "Here we show that the regulator of phosphokinases magnesium (Mg^2+^), exerts an enhancing effect on both the chemical and thermal sensitivity of TRPV2 endogenously expressed in rat bone marrow-derived macrophages."

– "Considering that TRPV2 is abundantly and functionally expressed in macrophages where other types of TRPV channels are barely detectable (Figure 1 —figure supplement 1) (Link et al., 103 2010; Nagasawa et al., 2007), we used rat bone marrow-derived macrophages (rBMDMs) as an endogenous cell system to record TRPV2 currents."

– "Our data show that Mg^2+^ modulates tyrosine phosphorylation levels of the TRPV2 channel protein, and thereby also its activity."

– "For instance, it has been reported that magnesium deficiency caused by genetic deficiencies in MAGT1 impairs anti-virus immune response".

– "the cell through ion channels such as TRPM6, TRPM7, or/and magnesium transporters like MagT1".

– "Using a TRPV2 mutant deficient in Mg^2+^ permeation and supplying the Mg^2+^-chelator EDTA through the patch-clamp glass pipette, our data suggest that transient Mg".

*Reviewer #2 (Recommendations for the authors):*

I am satisfied with most responses and modifications by the authors. I would, however, like to call attention to concern (8), rebuttal figure R3 and updated results on TRPV2 temperature dependence. The main text does not reflect the changes proposed by the rebuttal. Calculation of the temperature threshold for activation is still based on a 10% of maximal response (mentioned in Figure 3) and is not, as indicated in the Methods section ("Temperature Jump": line 536), done in accordance with the technique of Zhang et al. (2018). I appreciate that the example in R3 showed close agreement between their previous method and the preferred one that normalizes to a common temperature point. However, if their methods section claims the use of the optimal approach, then I expect it to be applied consistently.

A clear example that beckons standardizing the normalization to a common temperature can be seen in Figure 6 and the interpretation made from these data. Here the experimental temperature range in panels 6I-J is shifted in comparison to mock-treated cells.

Consequently, the summary plot in 6K shows a normalization to different final temperatures in each sample set, perhaps staggering the threshold temperatures of activation shown in 6L in the same order as the normalization temperature. To me, it is a serious concern that the arbitrary arrival at ~40 degrees C as the thermosensitive threshold in these data makes its way into the abstract of the report.

---

## [Author Response]

Essential revisions:Revisions requiring additional experimental work:1) The authors should include experiments at least in BMDM and TRPV2-expressing HEK293 cells in which 0.3 mM 2-APB is applied for the same duration as 0.3mM 2APB + magnesium. Ideally, this should have been done in all experiments, but I realize this would take a significant amount of effort and that given the rest of the evidence provided, it seems unlikely that this will affect the conclusions.

Following the suggestion, we have performed new experiments in BMDMs and TRPV2-expressing HEK293T cells. As illustrated in Figure 1 —figure supplement 3, prolonged application of 0.3 mM 2-APB alone didn’t have notable sensitizing effect on TRPV2 currents, while subsequent application of the same stimulus in the presence of 5 mM Mg^2+^ with same duration produced a significant increase of the TRPV2 currents. Hence, these new data are consistent with our previous observation and affirm our original conclusions. We now add this description in the main text (page 7, lines 813) and present the results in Figure 1 —figure supplement 3.

2) In figure 2 the authors demonstrate that Mg applied alone does not activate TRPV2 in the inside-out configuration (Figure 2G-H). The authors should include an inside-out experiment to address whether intracellularly applied 2-APB together with magnesium have a sensitizing effect.

Thanks for the insightful comments. We have conducted the suggested experiment. As shown in Figure 4—figure supplement 2, the inside-out recordings from TRPV2-expressing HEK293T cells at +60 mV show that the presence of Mg^2+^ increased the 2-APB response. However, the excised membrane patches might attach portion of tyrosine kinase JAK1. We then repeated the experiments by pre-treatment with the JAK1 inhibitor, Ruxolitinib, which indeed reduced the enhancement caused by Mg^2+^. This confirms the Mg^2+^-induced TRPV2 current enhancement is modulated by JAK1 phosphorylation.

In addition, we tested the effect of Mg^2+^ on inside-out recordings of

TRPV2(Y335/471/525F) mutant that loses the capability to be phosphorylated by JAK1. As expected, Mg^2+^ failed to enhance the 2-APB-evoked currents. Together, these data corroborate the Mg^2+^-JAK1-mediated phosphorylation contributes to the increased sensitivity of the TRPV2 channel. Of note, our findings also suggest that though exerting a facilitatory effect, Mg^2+^ alone is unable to activate TRPV2, and the excised membrane patches cannot completely isolate the regulatory effect of the intracellular signaling pathway occurring on underneath the cell membrane site.

3) Ion permeability of TRPV2 is not very well explored in previous reports. Thus, permeation of Mg through TRPV2 should be convincingly demonstrated by performing measurements of the reversal potential in different concentrations of extracellular magnesium.

As suggested, we conducted the new experiment to characterize the relative selectivity’s for Mg^2+^ through TRPV2 channel by measuring the reversal potential. For cation substitution experiment, a normal bath solution was used to establish a whole cell configuration. After the whole-cell configuration was obtained, bath solution was replaced with the solutions containing 110 mM Mg^2+^. As shown in Figure 2 —figure supplement 1, the measured *I-V* curves for different cationic solutions, and the zero-bias current indicates that Mg^2+^ ions pass more easily than Na^+^. We now add this description in Material and Methods (page 27, lines 8-14, and page 31, lines 38), and present the results in Figure 2 —figure supplement 1.

Other essential revisions, including experiments that are suggested, but not required for publication:4) The authors included ATP in the pipette in most of their experiments, which could be expected to be essential for observing the sensitizing effect of magnesium. This needs to be discussed. It would strengthen the manuscript if the authors included experimental data to directly address whether ATP in the pipette is required for observing the sensitizing effect of magnesium co-applied with 2-APB.

We previously showed that replacing cytosolic ATP with AMP-PNP, a nonhydrolyzable analog of ATP, eliminated the regulatory effect of Mg^2+^ (Figure 3C), indicating that ATP is required for the effect of Mg^2+^. According to the reviewer’s suggestion, we have carried out new experiments without the addition of ATP in the pipette solution. The sensitizing effect of Mg^2+^ was also observed in the whole-cell recordings from TRPV2-expressing HEK293T cells (Figure 3—figure supplement 1). This is most likely due to the abundance of ATP in the cell which is not rapidly diluted by the patch pipette solutions. Or the intracellular ATP has a concentration gradient at various sites and is associated with endogenous enzymes at different localizations.

5) The manuscript would benefit from additional editing to improve readability. A more concisely worded introduction that begins with the general properties of TRPV2 and then highlights our gaps in understanding, especially in any immunological role, would improve the transition into the summarized findings. The underlying hypothesis leading the authors to challenge TRPV2 with high concentrations of extracellular Mg should be (better) stated. There are many ways to explore the role of protein phosphorylation in the activity of ion channels, the authors applied many elegant techniques in this study. The starting point with extracellular Mg is somewhat odd, giving the slight impression that the authors did not have protein phosphorylation in mind when they started this study. The discussion should review and discuss the results in a manner consistent with the order of the figures. I would recommend discussing the Mg^2+^ dependent TRPV2 results first and the phosphorylation effect next.

Many thanks for the insightful suggestions. First, we update the introduction by pointing the necessity to understand the activity features of TRPV2 in immune cells, prior to transiting the study to mechanism studies (page 4, lines 4-5). Second, we place the early reports on potential effects of Mg^2+^ on TRP channels prior to the exploration of its role in TRPV2 channel regulation, so to link to the current experimental work (page 6, lines 5-6). Finally, we adjust the discussion first on the Mg^2+^ dependent TRPV2 potentiation and the phosphorylation effect next (page 17, lines 7-11 and lines 15-22, and page 18, lines 1-8).

6) In general, the wording in figure descriptions needs to be more concise and clearly relatable to figure legends. For example, Figure 1B indicates the response to washout but the figure description indicates responses to 0.3mM 2-APB with either 0 or 5 mM Mg^2+^. This discrepancy needs to be clarified. I would recommend small colored triangles (blue, red) in the current trace of Figure 1A, which indicate the time points at which the summary data measurement is obtained. This recommendation should be extended to all examples of summary plots that employ the same type of comparison (ie. Figure 1D, F, etc.).

Following the suggestion, we have revised the figure legends for Figure 1B and Figure 1D (page 41, lines 9-10 and page 42, lines 2-3), and also added small colored triangles to indicate the current amplitudes used for data analysis in the current traces of all Figures, when applicable.

7) The authors should clarify the duration of the exposure to 2APB and magnesium in all experiments where this could be an important factor affecting the results (see public review). The duration of stimulation should be the same or very similar at least for all key experiments – Figure 3E, Figure 3J and K, Figure 4A, G, H, and I, Figure 5B.

We agree that the duration of the stimulus might affect the final outcome. Due to the differences between cells, we sticked to the results only when the cell’s responses to stimulation can reach a steady state. By this functional standard, the application duration of 2-APB and magnesium in our experiments fall in comparable ranges (~70-100 s) for Figure 3E, Figure 3J and K, Figure 4A, G, H, and I, Figure 5B. Our results are consistent across experimental conditions.

8) All figures where current vs temperature relations at different conditions are compared should be normalized at the same temperature for all data, and the threshold calculated accordingly. Examples of this can be found in Zhang et al. (PNAS, 2017; https://www.pnas.org/doi/suppl/10.1073/pnas.1717192115).

Thank you for the comment. As suggested, we recalculated the temperature activation threshold by normalizing all data to the value derived at the same temperature. Take the bellowing Author response image 1 as an example, the temperature activation thresholds obtained by the new method are similar to our previous value, 52.6^o^C (new) vs. 53.2^o^C (previous) for control, and 45.3^o^C (new) vs. 46.7^o^C (previous) for being sensitized by Mg^2+^. The results are updated now in the main text, and we have also revised the statement on calculation of temperature activation threshold accordingly, and also cited the mentioned reference on page 28 line 13.

**Author response image 1. sa2fig1:** Determination of the temperature threshold for activation of TRPV2. (A) Representative current traces for TPRV2-expressing HEK293T cells in response to a family of temperature pulses ranging from 42 ^o^C to 57 ^o^C. (B) Current vs. temperature relations at −60 mV obtained from experiments as in (A). Individual cells are shown with currents normalized by their amplitude at 44 ^o^C. (C) Plot of log(I_test_/I_44_^_o_^_C_+1) obtained from the relations in (A). (D-F) Representative current traces, temperature-activation relations, and Plot of log(I_test_/I_44_^_o_^_C_+1) determinations for Mg^2+^ pretreated TRPV2expressing cells.

9) Line 15-16 on page 5. The authors state that the Mg-induced effect could not be washed out. Looking at the figures, however, it becomes clear that the washout period was only about 10s or so. This is much too short to say anything about if the Mg-induced sensitization of TRPV2 is reversible or not. This needs to be acknowledged and discussed.

Thanks for this comment. Indeed, if the washout period is extended, then the Mg^2+^-induced effect can be fully reversed (Author response image 2), further indicating that phosphorylation and dephosphorylation are always in dynamic equilibrium. Now we have removed this statement in the previous manuscript on page 5 lines 15-16.

**Author response image 2. sa2fig2:** Mg^2+^-induced sensitization of TRPV2 is reversible. (A) Representative whole-cell of TRPV2 currents evoked by 0.3 mM 2-APB, 0.3 mM 2-APB plus 5 mM Mg^2+^, subsequently repeated applications of 0.3 mM 2-APB and 3 mM 2-APB. (B) Time courses of peak currents elicited by repeated applications of 0.3 mM 2-APB after being sensitized by Mg^2+^. Currents were normalized to that evoked by 0.3 mM 2-APB before Mg^2+^ treatment.

10) On page 6, lines 9-12 the authors state that Mg is the only cation exhibiting a "more profound" effect on TRPV2. When looking at Figure 1 supplement 2 however, it appears other cations also have interesting effects on channel activity. The authors should comment on this.

Thank you for the careful reading. Indeed, as illustrated in Figure 1 supplement 2, a variety of different divalent cations exhibited different regulatory effects on TRPV2 currents. Among them, the enhancement effect of Mg^2+^ was the most prominent. Mg^2+^, Mn^2+^ and Co^2+^ enhanced the currents of TRPV2 to different degrees. By contrast, Ba^2+^, Cu^2+^ and Zn^2+^ had a remarkable inhibition of TRPV2 currents. Furthermore, the regulatory effects of different divalent cations on TRPV2 activity are worthy of an in-depth follow-up study. We have now added the comment to results (page 7, lines 17-20).

11) The authors imply that increased intracellular Mg-concentration accounts for channel phosphorylation. Why did the authors not perform experiments with different – pre-set – concentrations of Mg in the pipette solution? These experiments would be straightforward to perform, and they would make the approach with the TRPV2-E609/E614 mutant less critical.

The inclusion of Mg^2+^ in the pipette solution as shown in Figure 1—figure supplement 5 did enhance the sensitivity of the TRPV2 channel resulting in a leftward shift of the dose-response curve. However, we noticed that the addition of 5 mM Mg^2+^ to the pipette solution was less effective than the effect of Mg^2+^ influx from outside the cell through the channel. Possible reasons for this may lie in, (1) local transient concentration changes in Mg^2+^ are more capable of inducing enzymatic reactions, just as local ca^2+^ sparklets are more likely to facilitate subsequent reactions (Amberg et al., 2007); (2) a uniform global increase in Mg^2+^ concentration may trigger both phosphorylation and dephosphorylation, which may compromise the degree of TRPV2 phosphorylation.

12) The authors should remove the calcium imaging data and instead provide measurements of the reversal potential to show whether the mutations effectively reduce permeability to magnesium. Alternatively, all data for the E609/614Q should be removed from the manuscript.

Thank you for this comment, which is, related to comment 3. Figure 2 —figure supplement 1 exhibits the measured *I-V* curves for different cationic solutions, and the zero-bias current indicates that E609/E614Q reduced the permeability to Mg^2+^. Following the suggestion, we have replaced the ca^2+^-image data with the measurements of the reversal potential and also removed the description of ca^2+^-imaging in Materials and methods. We now add this description in the main text (page 9, line 12) and in Material and Methods (page 27, lines 8-14, and page 31, lines 3-8), and present the results in Figure 2 —figure supplement 1.

13) The authors should explain the smear that appears on the third lane in the in vitro kinase assay (Figure 3F).

JAK1-Flag proteins used in the in vitro kinase assay of this study were heterologously expressed in HEK293T cells and were immunoprecipitated with antiFlag agarose. Although the anti-Flag agarose was repeatedly washed to remove the proteins that interacted with JAK1, some of the interaction proteins remained and were phosphorylated during in vitro kinase assay, resulting in dispersion bands of varying sizes. To exclude the interference of other proteins, we prepared kinase-dead JAK1 (K908A) proteins. As shown in Figure 4E, there was no obvious phosphorylation signal when JAK1(K908A) was used in the in vitro kinase assay, suggesting that the kinase which phosphorylated TRPV2-Nt was JAK1.

Additionally, after obtaining the JAK1 proteins, there might be a small amount of degradation of JAK1 proteins due to the reaction time required in the in vitro kinase assay, and the smear was possibly the degradation of JAK1 proteins.

14) The authors should provide additional information on the mass-spectrometry results to highlight the relevant observations that support phosphorylation at Y335.

Sorry for the missed information. We have added additional information to the Materials and methods (page 29, lines 9-15) and figure legends (page 62, lines 2-9).

15) The authors should remove the phagocytosis assay from the manuscript, or alternatively, provide additional data with siRNA-dependent knockdown of TRPV2 channel expression to show specificity. SKF96365 is non-specific and more often employed to inhibit TRPC-channels. It may also block TRPV2. 100 µM SKF is a very high concentration that may even be cytotoxic. It would strengthen the manuscript if the authors provided whole-cell patch-clamp recordings of BMDM treated with TRPV2 siRNA to show that 2-APB responses are either absent or significantly reduced. Although the consistency between BMDM and HEK293 cell experiments gives confidence in the authors' conclusions, it should be considered that 2-APB is a highly non-specific agonist and that primary macrophages express other ion channels in addition to TRPV2.

Based on the results of qPCR, we found that the expression level of the *Trpv2* is the highest among all *Trpv* and *Trpc* channels (Figure 1 —figure supplement 1A). In addition, we also detected a higher abundance of expression of TRPV2 channel protein in BMDMs via immunoblotting (Figure 1 —figure supplement 1B).

We also performed new experiments by knocking down TRPV2 in BMDMs by shRNA (Figure 1 —figure supplement 1C). The whole-cell recordings showed that the cells significantly reduced their response to 2-APB stimulation (Figure 1 —figure supplement 1D-1F). To date, 2-APB, while not a specific agonist, has only been found to activate TRPV1, TRPV2, and TRPV3 channels, suggesting that the 2-APB-induced BMDMs currents are most likely from TRPV2 channels. Similar to the previous use of inhibitor SKF96365 that inhibited TRPV2 channel activity to reduce the phagocytic capacity of macrophages, knockdown of TRPV2 expression also reduced macrophage phagocytosis (revised Figure 3G). We now add this information in the main text (page 6, line 9) and the Materials and methods (page 24, lines 19-21), and present the results as Figure 1 —figure supplement 1.

16) The same concerns about the specificity of the findings apply to the current recordings with DRG neurons and BMDM – 2-APB is a non-selective agonist of many ion channel types, including TRP channels also expressed in DRG neurons. Ideally, the authors should have included DRG and BMDM recordings from TRPV2 KO animals, or from cells treated with siRNA. In the absence of this data, the authors should at least discuss these shortcomings.

Thank you for this comment, which is, related to comment 15. Based on the results of qPCR, we found that the TRPV1 and TRPV2 are highly expressed in DRG neurons, while the expression of other types of TRPV channels was barely detectable (Author response image 3). TRPV2 channels are predominantly expressed in medium- to large-sized DRG neurons that typically express fewer TRPV1 channels (Caterina et al., 1999). Additionally, we used capsaicin to confirm the lack of TRPV1 expression in recorded DRG neurons (Figure 1C). Although 2-APB is a non-selective agonist, has only been found to activate TRPV1, TRPV2, and TRPV3 channels so far. Thus, our use of 2-APB functionally targeted the TRPV2 ion channels.

**Author response image 3. sa2fig3:** Expression of TRPV2 in DRG neurons. Relative mRNA expression levels of different *Trpv* channels in BMDMs were assessed by qPCR.

17) The authors should provide additional discussion on the possible physiological implications of their findings. Importantly, the authors should discuss whether the magnesium dependence that they observe is consistent with physiological magnesium concentrations in the cytosol, and whether there are any known signaling pathways that could trigger the activity of JAK1 or PTPN1. The discussion about any Mg^2+^ dependent process as physiological seems overly speculative without additional experiments that maintain physiological conditions throughout experiments. It would be better to highlight the Mg^2+^ effect as an excellent tool to explore downstream modulation by cell signaling events. Any speculative aspect of the Mg^2+^ effect that is physiological should be clearly labeled as such and limited in scope.

The concentration of Mg^2+^ in the cytoplasm is about 10 to 30 mM, but most intracellular Mg^2+^ are bound to ribosomes, polynucleotides, and ATPs, resulting in the free Mg^2+^ concentration of about 0.3 to 1.2 mM (de Baaij et al., 2015; Funato et al., 2014; Moomaw and Maguire, 2008). We now add the information in the manuscript (page 17, lines 11-15).

To the best of our knowledge, there is only one study showing that Mg^2+^ activates PTPN1 (Bellomo et al., 2018) and no literature shows that Mg^2+^ can activate JAK1. Because in the previous studies the solution used for most in vitro kinase assay containing Mg^2+^, whether magnesium ion directly activates JAK1 has been ambiguous (Briscoe et al., 1996; Danial et al., 1998).

However, Mg^2+^ and JAK1 are well known for playing an important role in many physiological processes such as immune responses. Following the suggestions, we have revised the manuscript (page 19, lines 8-16).

18) The authors state that three residues (Y335, Y471, Y525) are conserved in other mammalian species. Human TRPV2 is known to often display clear species-specific differences as compared to rodent TRPV2. It would strengthen the manuscript if the authors provided data to address whether human TRPV2 channels are also subject to the regulatory mechanism described in the manuscript. However, this is not required and we leave it up to the authors to decide. Similarly, it would also be interesting if the authors examined whether this effect is conserved in TRPV1, TRPV3 and TRPV4 channels, but we realize this might be out of scope for the present manuscript.

Following the suggestion, we have carried out new experiments to study whether human TRPV2 channels are also subject to the regulatory mechanism. As illustrated below (Author response image 4), the Mg^2+^ shows the inhibition effect on human TRPV2. That might be due to Mg^2+^ blocking the human TRPV2 channel or Mg^2+^ simply inhibiting the single-channel current. Ion substitution experiments performed in whole-cell mode showed that the human TRPV2 channel is selective for Mg^2+^. Then Mg^2+^ which was dialyzed into the cell through recording pipette caused a left-shift of the concentration-response curve to 2-APB from the whole-cell experiments, and the EC_50_ of 2-APB on TRPV2 activation was shifted to 2.4 ± 0.2 mM from 3.4 ± 0.1 mM in the presence of 5 mM Mg^2+^. In addition, PTP inhibitor 2 also induced a left shift of the concentration-response curve to 2-APB. Together, these results suggest that human TRPV2 channels are also probably subject to the regulatory mechanism as described in this study, yet hTRPV2 current is likely to be inhibited by extracellular Mg^2+^.

Interestingly, it has been reported that extracellular Mg^2+^ potentiated rodent TRPV1 (Yang et al., 2014), while Lou J. and colleagues witnessed that rodent TRPV3 is inhibited by [Mg^2+^] (Luo et al., 2012).

**Author response image 4. sa2fig4:** Effect of Mg^2+^-dependent phosphorylation on human TRPV2. (A) Whole-cell currents at -60 mV in a hTRPV2-expressing HEK293T cell treated with 1.5 mM 2-APB, 1.5 mM 2-APB plus 5 mM Mg^2+^, and 3 mM 2-APB. (B) Relative cationic permeabilities (*P_X_*/*P_Na_*) of human TRPV2 channels assessed by reversal potentials. (C-E) Representative whole-cell currents evoked by increasing concentrations of 2-APB for human TRPV2-expressing HEK293T cells. The cells were under control contions (A), with addition of 10 mM Mg^2+^ in the pipette solution (B), and pre-treated with PTP inhibitor 2 (C). (F) Dose-response curves of 2-APB. Fitting by Hill’s equation resulted in the following: EC_50_ = 3.4 ± 0.1 mM and n_H_ = 2.7 ± 0.2 for control (*n* = 7); EC_50_ = 2.4 ± 0.2 mM and n_H_ = 2.7 ± 0.5 for addition of Mg^2+^ (*n* = 6) and EC_50_ = 1.9 ± 0.1 mM and n_H_ = 2.9 ± 0.5 for treatment by PTP inhibitor 2 (*n* = 6).

19) The 2-APB concentrations in the figure and the figure legend are not consistent in Figure5A.Figures 5A/6D: Explain why the concentration change to 0.03 mM/ 0.1 mM 2APB are necessary compared to earlier experiments (0.3 mM) and provide examples of a minimally effective phosphatase such as LB-100/ and shControl as representative traces.

Treatment with PTP inhibitors would elevate the basal phosphorylation levels of TRPV2 which caused a left-shift of the concentration-response curve to 2APB, which caused us to shift the concentration changes to 0.03 mM/ 0.1 mM 2-APB as presented in Figure 5A/6D. Due to the space constraints, the examples of LB-100 or shControl are not appended to the main figures. Instead, we show them here to provide the information for the editors and reviewers. As shown in Author response image 5, we observed that the cells that were treated with LB-100 or transfected with shControl elicited a small current by 0.3 mM 2-APB, which are consistent with our earlier experiments.

**Author response image 5. sa2fig5:** Effect of several phosphatase inhibitors for Mg^2+^-dependent enhancement of TRPV2. Whole-cell recordings from TRPV2-expressing HEK293T cell were consecutively challenged with 0.3 mM 2-APB, 0.3 mM 2-APB plus 5 mM Mg^2+,^ and 3 mM 2-APB. The cells were treated with LB-100

20) Page 2, line 3: it is unclear what the peripheral or central functions are. TRPV2 plays important roles in many non-neuronal tissues.

We have rephrased the sentence as: Transient receptor potential vanilloid 2 (TRPV2) is a multimodal ion channel implicated in diverse physiopathological processes (Page 2, lines 2-3).

21) The role of TRPV2 in thermo-, mechano- and osmo-sensing (page 3, lines 6-7 and 11-12) is controversial. The authors should acknowledge this in their introduction.

We now revise this information in the introduction (page 3, lines 15-18).

22) Page 4, lines 20-21: I suggest removing the statement about therapeutics. Alternatively, the authors should be more specific about what pathology they are referring to.

According to the suggestion, we have removed the statement about therapeutics.

23) Page 18, lines 1-2: I do not think that any of the findings in the manuscript indicate that long-term supplementation with magnesium can increase its cytosolic concentration.

It has been reported that the concentration of intracellular free Mg^2+^ can be increased by long-term Mg^2+^ supplementation (Chaigne-Delalande et al., 2013). In addition, Schmitz C and colleagues also found that supplementation of extracellular Mg^2+^ could rescue the viability and proliferation of Mg^2+^-deficient cells (Schmitz et al., 2003).

[Editors' note: further revisions were suggested prior to acceptance, as described below.]

Essential revisions:1) Current vs temperature relations in the manuscript figures are normalized differently between the channel constructs that are being compared. The threshold for heat activation appears to be calculated based on this same normalization scheme, which is different from what is indicated in the Methods section ("Temperature Jump": line 536), done in accordance with the technique of Zhang et al. (2018). We appreciate that the example in R3 showed close agreement between their previous method and the preferred one that normalizes to a common temperature point. However, if their methods section claims the use of the optimal approach, then this should be applied consistently in all figures displaying current-temperature relations.

Per suggestions, we have now updated all figures for displaying currenttemperature relations in accordance with the technique of Zhang et al. (2018) in new Figure 1I-N, Figure 5H-P, Figure 6F-K, and Figure 5 —figure supplement 1D-L. We also removed ‘to ~40 °C’ from the abstract to be more concise.

2) Please include zero-current lines for all current traces, so that readers can assess the magnitude of the leak currents in each recording.

Thanks for the suggestion, we have added zero-current lines for all current traces in all figures.

3) The new reversal potential measurements with external Na^+^ and Mg^2+^ show that whereas the E609/E614Q double mutant channel has lower Mg^2+^-permeability than WT, Mg^2+^ can still permeate through the mutant channel at least to the same extent as Na^+^ does. This seems difficult to reconcile with the strong negative effect that the double mutant has on the potentiation caused by external Mg^2+^ in Figure 2C and D. Two alternative explanations for the observed results that were not tested are that the E609/E614Q mutations prevent phosphorylation of key tyrosine residues, or that channels lose to capacity to become sensitized by tyrosine phosphorylation. Because these data can be confounding unless additional experimental work is included, we ask the authors to remove these data from the manuscript, or tone down the significance of these findings and mention alternative explanations for the observations that could not be ruled out by the data.

Thanks for the insightful comments. We have removed E609/614Q data from the manuscript as suggested. We also appreciate the alternative explanations provided by the reviewer. As for why the E609/E614Q double mutant channel showed negative effect to the external Mg^2+^, one possible reason might be due to the very different extracellular concentrations of the two ions, 5 mM Mg^2+^ vs. 145 mM Na^+^, used in the potentiation experiment. Though both ions have the similar permeability, there may be less Mg^2+^ entering the cell. That said, we fully agree that the effect of this double mutant channel deserves further exploring.

4) The data in Figures R1 (exposure of inside-out patches to Mg^2+^ and 2-APB applied together) and R2 (effect of ATP in the pipette) should be included in the manuscript as a supplement. These results add important information that is not conveyed elsewhere. The data in Figure R5 should also be included in the manuscript as a Supplementary Figure because it highlights an important methodological constraint, namely that the inclusion of Mg^2+^ in the pipette is much less effective in sensitizing TRPV2 channels than when applied from the extracellular side.

Following the suggestions, we now add Figure 4 —figure supplement 2, and add the description in the main text (page 14, lines 11-22, and page 15, lines 1-5). In addition, we add Figure 3 —figure supplement 1, and described in the main text (page 10, lines 22, and page 11, lines 1-5). We further add Figure 1 —figure supplement 5, and append the results in the main text (page 8, lines 3-6) and in the discussion (page 19, lines 9-15).

5) The apparent affinity for 2-APB seems to be increased in TRPV2 channels containing the triple mutation Y335/471/525F. This is contrary to what is expected from a mutation that mimics a non-phosphorylated state and runs counter to the main conclusions of the manuscript. This apparent discrepancy should be discussed.

Thanks for the careful reading. We now add “Notably, TRPV2(Y335/471/525F) was a little more sensitive to 2-APB. One possible reason is that the triple mutation might somehow alter the channel conformation and result in the increased sensitivity to its chemical agonist.” in the main text (page 14, lines 4-6).

6) The text contains several typos and stylistic errors that should be fixed. Here are some that I found, with suggested edits already incorporated:– "whether TRPV2 can function as a temperature sensor or a mechanical sensor in physiology still remains in debate."– "In mast cells, TRPV2-mediated calcium flux stimulates protein kinase A (PKA)-dependent proinflammatory degranulation (Stokes 68 et al., 2004)."– "Here we show that the regulator of phosphokinases magnesium (Mg^2+^), exerts an enhancing effect on both the chemical and thermal sensitivity of TRPV2 endogenously expressed in rat bone marrow-derived macrophages."– "Considering that TRPV2 is abundantly and functionally expressed in macrophages where other types of TRPV channels are barely detectable (Figure 1 —figure supplement 1) (Link et al., 103 2010; Nagasawa et al., 2007), we used rat bone marrow-derived macrophages (rBMDMs) as an endogenous cell system to record TRPV2 currents."– "Our data show that Mg^2+^ modulates tyrosine phosphorylation levels of the TRPV2 channel protein, and thereby also its activity."– "For instance, it has been reported that magnesium deficiency caused by genetic deficiencies in MAGT1 impairs anti-virus immune response".– "the cell through ion channels such as TRPM6, TRPM7, or/and magnesium transporters like MagT1".– "Using a TRPV2 mutant deficient in Mg^2+^ permeation and supplying the Mg^2+^-chelator EDTA through the patch-clamp glass pipette, our data suggest that transient Mg".

Many thanks for your kind help. All fixed. We have further verified the text and carefully revised the whole manuscript.